# Certified Unlearning for Neural Networks

Anastasia Koloskova [* 1]   Youssef Allouah [* 2]   Animesh Jha [1]   Rachid Guerraoui [2]   Sanmi Koyejo [1]

## Abstract

We address the problem of machine unlearning, where the goal is to remove the influence of specific training data from a model upon request, motivated by privacy concerns and regulatory requirements such as the "right to be forgotten." Unfortunately, existing methods rely on restrictive assumptions or lack formal guarantees. To this end, we propose a novel method for certified machine unlearning, leveraging the connection between unlearning and privacy amplification by stochastic post-processing. Our method uses noisy fine-tuning on the retain data, i.e., data that does not need to be removed, to ensure provable unlearning guarantees. This approach requires no assumptions about the underlying loss function, making it broadly applicable across diverse settings. We analyze the theoretical trade-offs in efficiency and accuracy and demonstrate empirically that our method not only achieves formal unlearning guarantees but also performs effectively in practice, outperforming existing baselines. Our code is available at https://github.com/stair-lab/certified-unlearning-neural-networks-icml-2025

## 1. Introduction

Machine unlearning—the process of removing the influence of specific training data from a model—has become an increasingly important challenge in modern machine learning (Nguyen et al., 2022). With the widespread adoption of deep learning in fields such as healthcare, natural language processing, and computer vision, concerns over data privacy, security, and control have grown significantly. In particular, regulatory frameworks like the General Data Protection Regulation (GDPR) of the European Union (Voigt & Von dem Bussche, 2017) enforce the *"right to be forgotten"*, requiring organizations to delete user data upon request. However, simply removing data from storage is insufficient if the information remains embedded in a trained model. This has led to the growing interest in unlearning techniques, which seek to eliminate the influence of specific data points while preserving the overall utility of the model. Achieving efficient and reliable unlearning is particularly challenging when working with large-scale neural networks, where full retraining from scratch is computationally prohibitive.

The idea of machine unlearning dates back to Cao & Yang (2015) and has since inspired a range of approaches. Broadly, unlearning techniques fall into two categories: *exact unlearning*, which aims to completely erase the influence of specific data points, and *approximate unlearning*, which seeks a computationally efficient but approximate removal of information. While exact unlearning offers the strongest theoretical guarantees, it is rarely practical for large-scale models or frequent unlearning requests due to its prohibitive computational costs (Ginart et al., 2019; Bourtoule et al., 2021). As a result, many existing methods adopt relaxed guarantees, but these often come with significant trade-offs. For instance, some approaches rely on restrictive assumptions about loss functions, such as convex linear models (Guo et al., 2020), while others lack rigorous theoretical guarantees (Graves et al., 2021; Kurmanji et al., 2024) or require extensive retraining (Bourtoule et al., 2021). One common heuristic method is fine-tuning on retained data, and potentially gradient ascent on the forget data, to induce catastrophic forgetting in the affected parts of the model (Triantafillou et al., 2024). While this technique has shown promise in reducing the retained influence of forgotten data, it does not inherently provide certifiable guarantees of unlearning, leaving open questions about its reliability in privacy-sensitive applications.

To address these limitations, we propose a new approximate unlearning framework that builds on the concept of privacy amplification by stochastic post-processing (Balle et al., 2019). Our method leverages noisy fine-tuning on retained data to enforce provable unlearning guarantees while maintaining computational efficiency. Unlike existing approaches, our framework does not impose restrictive assumptions on the loss function, making it particularly

*Equal contribution   [1]Stanford University, USA   [2]EPFL, Switzerland. Correspondence to: Anastasia Koloskova <anakolos@stanford.edu>, Youssef Allouah <youssef.allouah@epfl.ch>.

*Proceedings of the 42nd International Conference on Machine Learning*, Vancouver, Canada. PMLR 267, 2025. Copyright 2025 by the author(s).

well-suited for non-convex optimization problems such as deep learning. We interpret each noisy fine-tuning step as a form of stochastic post-processing, ensuring that privacy improves progressively with each iteration while balancing the trade-offs between accuracy and computational cost. Through rigorous theoretical analysis and extensive empirical validation, we demonstrate that our method provides both formal guarantees and strong practical performance, making it a viable solution for real-world machine learning applications where frequent unlearning requests must be handled efficiently.

Our key contributions can be summarized as follows:

- A novel certified unlearning method that integrates noisy fine-tuning with privacy amplification by stochastic post-processing, offering a principled approach to approximate unlearning.

- Rigorous unlearning guarantees that do not depend on restrictive assumptions such as loss function smoothness, making the method applicable to a broad range of models, including deep neural networks.

- Empirical validation in deep learning applications, demonstrating that our approach not only meets formal unlearning guarantees but also surpasses existing baselines in performance and model utility.

### 1.1. Related Work

**Amplification by post-processing.** Our approach is inspired by privacy amplification via stochastic post-processing, a concept in differential privacy where randomized transformations that do not use private data enhance privacy guarantees. The foundational work of Feldman et al. (2018) introduced privacy amplification by iteration, demonstrating that when training with differentially private stochastic gradient descent (DP-SGD) on convex objectives, the privacy of an unused training sample improves with the number of optimization steps. Balle et al. (2019) extended this result, refining the analysis of amplification by iteration and introducing amplification by mixing, which establishes that privacy is further strengthened when applying a Markov kernel satisfying specific mixing conditions. Subsequent work by Asoodeh et al. (2020) showed that under bounded domain assumptions, these mixing conditions hold for the Gaussian mechanism, leading to tighter privacy guarantees for DP-SGD. Our proposed unlearning framework—based on noisy fine-tuning on retained data—operates as a stochastic post-processing step that does not access the data to be forgotten. Thus, we extend privacy amplification techniques beyond convex settings to enable certified unlearning in deep learning models.

**Certified unlearning.** There is a growing body of work on certified unlearning, but existing approaches are largely inapplicable to neural networks. Most prior methods rely on restrictive assumptions about the model or loss function that do not hold for deep learning.

Several works focus on convex tasks (Guo et al., 2020; Neel et al., 2021; Sekhari et al., 2021; Allouah et al., 2025), but this limits their applicability to deep learning. These methods work in practice for logistic regression-style tasks and do not extend to neural networks due to non-convexity.

Recent works aim to achieve certified unlearning for non-convex tasks (Golatkar et al., 2020; Chourasia & Shah, 2023; Chien et al., 2024; Mu & Klabjan, 2024; Zhang et al., 2024; Allouah et al., 2025), but still impose significant constraints. All require the loss function to be smooth, limiting them to networks with smooth activations. Most (Chourasia & Shah, 2023; Chien et al., 2024; Mu & Klabjan, 2024) also require knowledge of the smoothness constant, restricting applicability to simpler models where this constant is tractable. Furthermore, Allouah et al. (2025) additionally assumes a unique minimizer, excluding virtually all practical neural architectures. Zhang et al. (2024) additionally requires knowledge of the minimal eigenvalue of the Hessian at the unique optimal model. To the best of our knowledge, our approach is the first certified unlearning method that supports arbitrary non-convex tasks, enabling provable unlearning guarantees for practical deep learning models.

**Unlearning applications.** A separate line of research focuses on concept unlearning, which aims to remove specific topics or themes from language models, beyond forgetting particular training samples (Liu et al., 2024). For instance, work in this domain has explored techniques for eliminating knowledge of topics like "Harry Potter" or other potentially harmful or unethical content from large language models (Eldan & Russinovich, 2023). These methods often involve intervention at the representation or knowledge distillation level, rather than enforcing formal guarantees of data removal. In contrast, our work focuses on data point unlearning, ensuring that information associated with specific training samples is provably removed while preserving model utility. Other explored unlearning settings include unlearning in graph neural networks (Chien et al., 2022), in min-max optimization settings (Liu et al., 2023), and the adversarial setting with the server possibly forging unlearning (Thudi et al., 2021).

## 2. Problem Statement

We consider a model $\hat{\mathbf{x}} \in \mathbb{R}^d$ trained using algorithm $\mathcal{A}$ on a dataset $\mathcal{D}$ of $n$ training examples, i.e., $\hat{\mathbf{x}} = \mathcal{A}(\mathcal{D})$. We place no restrictions on $\mathcal{A}$; it may be SGD, Adam, momentum-SGD, etc. An unlearning request specifies a subset $\mathcal{D}_f \subset \mathcal{D}$, referred to as the forget set, which we wish to erase from the model. Ideally, we could retrain the model from scratch on

the retain set $\mathcal{D} \setminus \mathcal{D}_f$, yielding $\mathbf{x}_u = \mathcal{A}(\mathcal{D} \setminus \mathcal{D}_f)$, but this is often computationally prohibitive. Instead, we aim to design an approximate unlearning algorithm $\mathcal{U}$ that outputs a model retaining "no information" about $\mathcal{D}_f$. To this end, $\mathcal{U}$ takes as input the original model $\hat{\mathbf{x}} = \mathcal{A}(\mathcal{D})$, the unlearning request $\mathcal{D}_f$, and the full training dataset $\mathcal{D}$. Formally, unifying several prior definitions (Ginart et al., 2019; Guo et al., 2020), we require $\mathcal{U}$ to satisfy the guarantees below.

**Definition 2.1** (($\varepsilon, \delta$)-unlearning). Let $\varepsilon \geq 0, \delta \in [0, 1]$. We say that $\mathcal{U}$ is ($\varepsilon, \delta$)-unlearning algorithm for $\mathcal{A}$ if there exists a certifying algorithm $\bar{\mathcal{A}}$, such that for any forget and initial datasets $\mathcal{D}_f \subset \mathcal{D}$ and any observation $O \in \mathbb{R}^d$,

$$\Pr[\mathcal{U}(\mathcal{A}(\mathcal{D}), \mathcal{D}, \mathcal{D}_f) = O] \leq e^\varepsilon \Pr[\bar{\mathcal{A}}(\mathcal{D} \setminus \mathcal{D}_f) = O] + \delta,$$
$$\Pr[\bar{\mathcal{A}}(\mathcal{D} \setminus \mathcal{D}_f) = O] \leq e^\varepsilon \Pr[\mathcal{U}(\mathcal{A}(\mathcal{D}), \mathcal{D}, \mathcal{D}_f) = O] + \delta.$$

For simplicity, we refer to approximate unlearning as ($\varepsilon, \delta$)-unlearning for some values of $\varepsilon$ and $\delta$, inspired by ($\varepsilon, \delta$)-differential privacy (Dwork & Roth, 2014). This formulation parallels differential privacy by treating $\mathcal{D}_f$ as "private" and $\mathcal{D} \setminus \mathcal{D}_f$ as "public," thereby ensuring ($\varepsilon, \delta$)-privacy for the forget set. Indeed, this notion ensures it is statistically difficult to distinguish the output of the unlearning algorithm $\mathcal{U}$ from that of a certifying algorithm $\bar{\mathcal{A}}$ that has no access to the forget set $\mathcal{D}_f$.

Importantly, the definition does not fix $\bar{\mathcal{A}}$ but only requires its existence, allowing flexibility to capture various prior definitions. For example, setting $\bar{\mathcal{A}} = \mathcal{A}$ recovers definitions from (Ginart et al., 2019; Guo et al., 2020), while setting $\bar{\mathcal{A}} = \mathcal{U}(\mathcal{A}(\mathcal{D} \setminus \mathcal{D}_f), \mathcal{D} \setminus \mathcal{D}_f, \varnothing)$ aligns with (Allouah et al., 2025; Sekhari et al., 2021). We primarily adopt the latter form in this work. Note that our choice of the certifying algorithm $\bar{\mathcal{A}}$ is purely theoretical and does not require running additional computational steps in practice. Finally, we emphasize that Definition 2.1 naturally supports non-adaptive sequential unlearning, where data points are removed one-by-one without revisiting earlier removals.

**Baselines.** We now describe two straightforward baselines for achieving ($\varepsilon, \delta$)-unlearning. The first baseline is *output perturbation*, which applies the standard Gaussian mechanism to the original model $\hat{\mathbf{x}}$. The procedure involves clipping the model parameters to ensure bounded sensitivity and adding Gaussian noise to the model's output. Formally, the unlearning procedure is defined as:

$$\mathbf{x}_0 = \Pi_{C_0}(\hat{\mathbf{x}}) + \boldsymbol{\xi}_0; \quad \boldsymbol{\xi}_0 \sim \mathcal{N}\left(0, \frac{8C_0^2 \ln(1.25/\delta)}{\varepsilon^2}\mathbf{I}_d\right), \tag{1}$$

where $\Pi_{C_0}$ represents the clipping operation, and $\boldsymbol{\xi}_0$ is noise sampled from the Gaussian distribution, with sufficient magnitude to ensure privacy (Dwork & Roth, 2014). While theoretically sound, output perturbation often performs poorly

in practice, as the required noise magnitude is large, which can significantly degrade the utility of the model.

Another baseline is *retraining from scratch*, where the forget dataset $\mathcal{D}_f$ is discarded, and the model is retrained from scratch on the remaining data $\mathcal{D} \setminus \mathcal{D}_f$ using the algorithm $\mathcal{A}$. Although this guarantees perfect unlearning (i.e., $(0, 0)$-unlearning), it is computationally expensive and requires substantial memory resources, which undermines the efficiency objectives of approximate unlearning.

## 3. Algorithm

Our approach is based on fine-tuning the model with stochastic gradient descent (SGD) using only the retained data $\mathcal{D} \setminus \mathcal{D}_f$, while incorporating regularization. Recall that in this approach, we initialize from the original model $\hat{\mathbf{x}}$ and update it for $T \geq 1$ iterations as follows for every $t \in \{0, \ldots, T-1\}$:

$$\mathbf{x}_{t+1} = \mathbf{x}_t - \gamma(\mathbf{g}_t + \lambda\mathbf{x}_t), \qquad \mathbf{x}_0 = \hat{\mathbf{x}}, \tag{2}$$

where $\mathbf{g}_t$ is the stochastic gradient computed on the retained data, $\gamma$ is the learning rate, and $\lambda$ a regularization parameter.

While this method may induce some empirical forgetting due to the phenomenon of catastrophic forgetting in SGD (Goodfellow et al., 2013), it does not provide formal ($\varepsilon, \delta$)-unlearning as required by Definition 2.1. Therefore, to frame each fine-tuning step as a stochastic post-processing operation (Balle et al., 2019) applied to the initial model $\hat{\mathbf{x}}$, we introduce *gradient clipping* and *model clipping*, both combined with Gaussian privacy noise. These modifications allow us to map the process to different stochastic post-processing mechanisms (Balle et al., 2019).

**Gradient clipping.** Our primary approach relies on gradient clipping, where we clip gradients before applying updates, followed by noise addition. This method resembles standard DP-SGD (Abadi et al., 2016) but differs in that gradient updates exclude any "private" (forget) data points:

$$\begin{aligned}\mathbf{x}_0 &= \Pi_{C_0}(\hat{\mathbf{x}}), \\ \mathbf{x}_{t+1} &= \mathbf{x}_t - \gamma\left(\Pi_{C_1}(\mathbf{g}_t) + \lambda\mathbf{x}_t\right) + \boldsymbol{\xi}_{t+1},\end{aligned} \tag{3}$$

where $\boldsymbol{\xi}_{t+1} \sim \mathcal{N}(0, \sigma^2\mathbf{I}_d)$ is Gaussian noise and $\Pi_{C_0}, \Pi_{C_1}$ are the clipping operators of radius $C_0 > 0$ and $C_1 > 0$ respectively[1]. We analyze this method in the theoretical framework of privacy amplification by iteration (Feldman et al., 2018), to show that redistributing noise across multiple steps enables a reduction in noise at the initial step while maintaining strong privacy guarantees. This noise reduction retains the original model's performance. The regularization with parameter $\lambda$ additionally allows implicitly

---

[1]Following standard differential privacy mechanisms, our clipping is by norm, i.e., $\Pi_C(\mathbf{x}) := \mathbf{x}\min\{\frac{C}{\|\mathbf{x}\|}, 1\}$ with radius $C$.

controlling the norm of the model $\mathbf{x}_t$ that is increased due to the presence of the noise $\boldsymbol{\xi}_t$.

**Model clipping.** An alternative approach involves model clipping, where each update is clipped to a predefined radius before noise addition:

$$
\begin{aligned}
\mathbf{x}_0 &= \hat{\mathbf{x}} \\
\mathbf{x}_{t+1} &= \Pi_{C_2}(\mathbf{x}_t - \gamma(\mathbf{g}_t + \lambda\mathbf{x}_t)) + \boldsymbol{\xi}_{t+1},
\end{aligned}
\tag{4}
$$

where $\boldsymbol{\xi}_{t+1} \sim \mathcal{N}(0, \sigma^2\mathbf{I}_d)$ and $\boldsymbol{\xi}_0 \sim \mathcal{N}(0, \sigma_0^2\mathbf{I}_d)$. Since the gradient $\mathbf{g}_t$ is computed solely on the retain data, the argument of the clipping operator can be interpreted as a post-processing transformation of the private model, given by the mapping $\psi(\mathbf{x}_t) := \mathbf{x}_t - \gamma(\mathbf{g}_t + \lambda\mathbf{x}_t)$. The clipping and noise addition ensure differential privacy guarantees (Dwork & Roth, 2014). Standard results on post-processing state that such transformations preserve or improve existing differential privacy guarantees. Additionally, privacy amplification by stochastic post-processing (Balle et al., 2019) suggests that each additional step can further enhance privacy beyond that of the previous model.

# 4. Theoretical Analysis

We now present the approximate unlearning guarantees of the gradient and model clipping variants of our unlearning method introduced in the previous section. We also theoretically compare these with previous non-convex certified unlearning algorithms. We defer all proofs to Appendix A.

## 4.1. Unlearning Guarantees

We first present in Theorem 4.1 the unlearning guarantees of the Gradient clipping approach, as defined in Equation (3).

**Theorem 4.1** (Gradient clipping). *Let $T \geq 1, \gamma, \sigma > 0, \lambda \geq 0, \delta \in (0, 1), \varepsilon \in (0, 3\log(1/\delta))$. Consider $T$ iterations of the unlearning algorithm defined in (3). We obtain $(\varepsilon, \delta)$-unlearning if:*

1. *Without regularization ($\lambda = 0$):*

$$
\sigma^2 = \frac{9\log(1/\delta)}{\varepsilon^2 T}(C_0 + C_1\gamma T)^2.
\tag{5}
$$

2. *With regularization ($\lambda > 0$): if $\gamma\lambda \in (\frac{1}{2}, 1)$ and*

$$
\sigma^2 = \frac{72\gamma\lambda\log(1/\delta)}{\varepsilon^2}\left(C_0(1 - \gamma\lambda)^T + \frac{C_1}{\lambda}\right)^2.
\tag{6}
$$

The proof relies on techniques from privacy amplification by iteration introduced by Feldman et al. (2018), which is a form of privacy amplification through stochastic post-processing (Balle et al., 2019). We extend the original analysis Feldman et al. (2018) that is applicable only to convex

functions to the non-convex case. The key ingredient in our analysis is the shift-reduction lemma (Lemma A.6), which enables us to control the growth of the Rényi divergence between (i) the unlearned model initialized from the full model and (ii) the unlearned model initialized from training only on the retained data, across iterations.

Unlike the original privacy amplification by iteration framework, which assumes smoothness and convexity of the loss function (Feldman et al., 2018), our approach circumvents these assumptions by introducing gradient clipping. This is reflected in the sufficient noise magnitude required to achieve $(\varepsilon, \delta)$-unlearning, as given in (5). Specifically, in the absence of regularization $\lambda = 0$, gradient clipping at threshold $C_1$ induces a dependence of the form:

$$
\sigma^2 = \mathcal{O}\left(\frac{\log(1/\delta)}{\varepsilon^2}\left(\frac{C_0^2}{T} + \gamma^2 T C_1^2\right)\right).
\tag{7}
$$

Intuitively, for a large enough number of iterations, we can trade off the effect of the initial clipping radius $C_0$ for a minor cost proportional to the learning rate $\gamma$ and the per-iteration clipping radius $C_1$. Fortunately, this cost can be controlled by choosing a sufficiently small learning rate $\gamma$. In particular, if $\gamma = \frac{C_0}{C_1 T}$, then asymptotically in $T$ we get $\sigma^2 = \mathcal{O}\left(\frac{C_0^2 \log(1/\delta)}{\varepsilon^2 T}\right)$. This implies that the required noise magnitude decreases as the number of iterations $T$ increases, ultimately tending to zero in the limit $T \to \infty$.

Given $C_0, C_1$, and $\gamma$, we can upper bound the optimal number of unlearning steps by minimizing (7). Specifically, setting $T$ larger than

$$
T^\star := \arg\min_T \left\{\frac{C_0^2}{T} + \gamma^2 T C_1^2\right\} = \frac{C_0}{\gamma C_1}
$$

leads to more iterations ($T > T^\star$) with increased noise per iteration ($\sigma_T \geq \sigma_{T^\star}$ by definition of $T^\star$), while achieving the same $(\epsilon, \delta)$-unlearning.

Finally, in the regularized case $\lambda > 0$, the noise expression (6) for achieving $(\varepsilon, \delta)$-unlearning simplifies to:

$$
\sigma^2 = \mathcal{O}\left(\frac{\gamma\log(1/\delta)}{\varepsilon^2}\left(\lambda C_0^2 \exp(-\lambda\gamma T) + \frac{C_1^2}{\lambda}\right)\right).
\tag{8}
$$

Here, regularization enables an exponential reduction in $T$ of the dependence on the initial clipping threshold $C_0$, effectively mitigating its impact over time. However, this comes at the cost of an increased dependence on the per-iteration clipping threshold $C_1$, scaling inversely with the regularization factor $\lambda$. While this suggests a smaller noise magnitude per iteration, overly strong regularization may degrade the model's performance, as we further analyze in the next section. Finally, we provide a refined, but complex, formula for noise magnitudes in Theorem A.9 (appendix)

using Rényi divergences, which we apply before precise Rényi-to-DP conversion (Balle et al., 2020) in practice.

Next, we state the unlearning guarantees of the Model clipping approach, as defined in Equation (4).

**Theorem 4.2** (Model clipping). *Let $T \geq 1, C_0, C_2, \sigma_0, \varepsilon > 0$, and $\delta \in (0,1)$. Denote for every $r > 0$,*

$$\theta_\varepsilon(r) := Q\left(\frac{\varepsilon}{r} - \frac{r}{2}\right) - e^\varepsilon Q\left(\frac{\varepsilon}{r} + \frac{r}{2}\right), \qquad (9)$$

*where for all $t \in \mathbb{R}$, $Q(t) := \frac{1}{\sqrt{2\pi}} \int_t^\infty e^{-u^2/2} du$.*

*Consider $T$ iterations of the unlearning algorithm defined in* (4). *We obtain $(\varepsilon, \delta)$-unlearning if*

$$T \geq \frac{\log(1/\delta) + \log \theta_\varepsilon(\frac{2C_0}{\sigma_0})}{\log\left(1/\theta_\varepsilon(\frac{2C_2}{\sigma})\right)}. \qquad (10)$$

*In particular, for any $T \geq 1, \varepsilon \in (0,1)$, it suffices to have*

$$\sigma^2 = \frac{8C_2^2 \ln(1.25)}{\varepsilon^2}\left[1 + \frac{1}{T}\left(\ln(1.25/\delta) - \frac{\sigma_0^2\varepsilon^2}{8C_0^2}\right)\right]. \qquad (11)$$

The proof of Theorem 4.2 relies on recent advances in privacy amplification by stochastic post-processing due to Balle et al. (2019) and Asoodeh et al. (2020). The initial iteration in (4) provides $(\varepsilon_0, \delta_0)$ unlearning since it is the standard Gaussian mechanism from differential privacy (Dwork & Roth, 2014). Moreover, assuming that the model at step $t$ of Algorithm (4) guarantees $(\varepsilon_t, \delta_t)$-unlearning, using the contraction coefficients (Asoodeh et al., 2020) approach, we show that the next iteration amplifies the unlearning guarantee as follows:

$$(\varepsilon_{t+1}, \delta_{t+1}) = (\varepsilon_t, \theta_\varepsilon(\tfrac{2C_2}{\sigma}) \cdot \delta_t),$$

where the expression of the amplification factor $\theta_\varepsilon(\frac{2C_2}{\sigma}) \in (0,1)$ is given in (9). This means that after $T$ iterations, our algorithm guarantees $(\varepsilon_T, \delta_T) = (\varepsilon_0, \theta_\varepsilon(\frac{2C_2}{\sigma})^T \delta_0)$ unlearning. Stated differently, given any target $\varepsilon, \delta$ and any noise magnitudes $\sigma^2, \sigma_0^2$ and clipping thresholds $C_1, C_0$, we show that it sufficient for the number of iterations to be at least that in (10) to guarantee $(\varepsilon, \delta)$-unlearning.

While the expression of the amplification factor $\theta_\varepsilon(\frac{2C_2}{\sigma})$ given in (9) is complex, we remark that it is a decreasing function of $\sigma$ taking values in $(0,1)$. In fact, assuming that $\varepsilon \in (0,1)$, and given any number of iterations $T$, we state a simple expression of the sufficient noise magnitude $\sigma^2$ in (11). This simplified expression is only for analytical purposes, since it gives looser unlearning guarantees.

### 4.2. Theoretical Comparison

To compare the various unlearning methods, we analyze the number of iterations required and the noise injected per iteration to achieve the same $(\varepsilon, \delta)$-unlearning guarantee. An effective method should minimize noise per iteration while keeping the number of iterations reasonable to preserve model accuracy. Since all methods rely on noisy updates, we focus on the magnitude of noise injected. A summary of our findings is provided in Table 1, with details below.

**Output perturbation.** We recall that this is a natural baseline, defined in (1), whereby we first project the original model with clipping threshold $C_0$ and add noise of magnitude $\sigma^2 = \frac{8\ln(1.25/\delta)C_0^2}{\varepsilon^2}$, which guarantees $(\varepsilon, \delta)$-unlearning for $\varepsilon, \delta \in (0,1)$, following (Dwork & Roth, 2014, Theorem A.1).

**DP training.** Training with DP guarantees unlearning for free; we do not need to inject noise after receiving unlearning requests. However, the noise needed may be larger than with other unlearning methods. Indeed, consider unlearning a set of $k$ samples with DP–SGD. For this, we need *group-DP*: if a mechanism is $(\varepsilon, \delta)$-DP for single-record changes, then it is $(k\varepsilon, \ ke^{k\varepsilon}\delta)$-DP for any pair of datasets that differ in $\leq k$ records (Vadhan, 2017, Lemma 2.2). Consequently, to attain the same $(\varepsilon, \delta)$-unlearning guarantee one must run DP–SGD with noise $\sigma^2 = \frac{2C^2k^2}{\varepsilon^2}\left(\ln(\frac{1.25k}{\delta}) + k\varepsilon\right)$. This is typically much larger noise than with our techniques, given that it is at least quadratic in $k$, which may scale with the size of the dataset. This is in line with recent findings on the theoretical separation between DP and certified unlearning (Sekhari et al., 2021; Allouah et al., 2025).

**Gradient clipping.** From Theorem 4.1, $T \geq 1$ iterations of Gradient clipping (3) with noise magnitude $\sigma^2 = \frac{9\log(1/\delta)}{\varepsilon^2 T}(C_0 + C_1\gamma T)^2$ satisfies $(\varepsilon, \delta)$-unlearning assuming $\varepsilon \leq 3\log(1/\delta)$. Setting $T = \frac{C_0}{\gamma C_1}$ minimizes the noise to

$$\sigma^2 = \frac{36\gamma C_1 C_0 \log(1/\delta)}{\varepsilon^2}. \qquad (12)$$

This substantially reduces noise per iteration compared to output perturbation—by a factor of $\frac{C_0}{C_1\gamma}$, which is significant when $\gamma \ll \frac{C_0}{C_1}$. A small learning rate or large initial clipping $C_0$ can make this method particularly effective in preserving model accuracy.

For the regularized version of gradient clipping we recall that noisy gradient descent with $T$ iterations and constant noise level, given by the expression $\sigma^2 = \frac{72\gamma\lambda\log(1/\delta)}{\varepsilon^2}\left(C_0(1-\gamma\lambda)^T + \frac{C_1}{\lambda}\right)^2$ satisfies $(\varepsilon, \delta)$-unlearning under the assumption $\varepsilon \leq 3\log(1/\delta)$. Setting $T = \frac{1}{\eta\lambda}\log(\frac{\lambda C_0}{C_1})$ of

$$\sigma^2 = \frac{C_1^2\gamma\log(1/\delta)}{\lambda\varepsilon^2}. \qquad (13)$$

This approach outperforms the unregularized variant when $\lambda > \frac{C_1}{C_0}$, requiring only logarithmic iterations in the initial

| Algorithm | Variance of Noise Injected | | Assumptions |
|---|---|---|---|
| | Max. per Iteration | Iterations | |
| Output Perturbation (baseline) | $C_0^2$ | 1 | |
| Gradient Clipping (3) | $\gamma C_1 C_0$ | $C_0/\gamma C_1$ | |
| Gradient Clipping (3) (w/ regularization) | $\gamma C_1^2/\lambda$ | $\log\left(\lambda C_0/C_1\right)/\gamma\lambda$ | |
| Model Clipping (4) | $C_2^2$ | $\log(1/\delta)$ | |
| Langevin Diffusion (Chourasia & Shah, 2023) | *no explicit expression* | – | smoothness, boundedness, noisy training |
| Rewind-to-Delete (Mu & Klabjan, 2024) | *exponential in smoothness constant* | – | smoothness, noisy training |

*Table 1.* Summary comparison of certified unlearning accountants for non-convex tasks. $C_0$: initial clipping threshold, $C_1/C_2$: running clipping threshold for gradient and model clipping resp., $\gamma$: learning rate, $\lambda$: $\ell_2$-regularization factor. We ignore absolute constants and multiplicative factor $\frac{\log(1/\delta)}{\varepsilon^2}$ which is in the noise variance of all methods. We note that the entry "Langevin Diffusion" also covers the work of Chien et al. (2024). *Model Clipping* and *Gradient Clipping* algorithms effectively reduce the maximum noise per iteration at the cost of doing more noisy SGD steps compared to the output perturbation. More details on the comparison are given in Section 4.2.

clipping radius $C_0$. This suggests that projecting onto a larger set can better preserve accuracy, though it may require stronger regularization, which could degrade performance in some tasks.

**Model clipping.** We recall from Theorem 4.2 that $T \geq 1$ iterations of Model clipping (4) with noise magnitude

$$\sigma^2 = \frac{8C_2^2 \ln(1.25)}{\varepsilon^2} + \frac{8C_2^2 \ln(1.25)}{T\varepsilon^2}\left[\ln(1.25/\delta) - \frac{\sigma_0^2\varepsilon^2}{8C_0^2}\right]$$

is provably sufficient to achieve $(\varepsilon, \delta)$-unlearning. Assume that the initial noise magnitude $\sigma_0^2$ is at most $\frac{8C_0^2 \ln(1.25/\delta)}{\varepsilon^2}$, as the latter magnitude is sufficient to obtain $(\varepsilon, \delta)$-unlearning in one iteration as in the output perturbation baseline. Therefore, the order of magnitude of the minimum value of the noise

$$\sigma^2 = \frac{8C_2^2 \ln(1.25)}{\varepsilon^2} \tag{14}$$

can be attained within $T = \ln(1.25/\delta)$ iterations. This represents a significant improvement over the baseline, reducing the noise per iteration by a factor of $\frac{C_0^2}{C_2^2}$. If $C_2$ is small or if the initial clipping threshold $C_0$ is aggressive, this reduction can be substantial. Compared to amplification by iteration, this method requires fewer iterations (only logarithmic in $1/\delta$), though it may introduce more noise per iteration when the learning rate is small.

**Prior works.** Existing certified unlearning methods that do not assume convexity of the loss function include (Chourasia & Shah, 2023; Chien et al., 2024; Mu & Klabjan, 2024). However, unlike our approach, these methods rely on the assumption that the loss function is smooth[2]

and are tailored to specific training algorithms. For instance, Chourasia & Shah (2023) and Chien et al. (2024) analyze training with noisy projected gradient descent, leveraging both smoothness and specific training dynamics to establish unlearning guarantees. These guarantees stem from the convergence of the training process to a limiting distribution, but additional restrictive assumptions are required. Notably, the smoothness constant is needed not only for theoretical analysis but also to determine the appropriate noise level at each step to achieve $(\epsilon, \delta)$-unlearning. This constraint limits the applicable function class to those where the smoothness constant is easily computable, effectively excluding modern neural networks. Chourasia & Shah (2023) assume that the loss function is bounded, while Chien et al. (2024) require that the training's limiting distribution satisfies an isoperimetric inequality. These constraints significantly limit the applicability of their methods, primarily to smooth convex tasks such as logistic regression (Chien et al., 2024). Similarly, Mu & Klabjan (2024) address non-convex loss functions but assume that training follows gradient descent with output perturbation. Their approach also relies on smoothness and requires injecting noise with a magnitude that scales exponentially with the smoothness parameter, which can be prohibitive in practice. In contrast, our approach removes these smoothness constraints and is not tied to a specific training algorithm, making it applicable to a broader class of learning problems.

## 5. Experimental Evaluation

In this section, we present an empirical evaluation of our proposed unlearning method, in its two variants Gradient Clipping (3) and Model Clipping (4), on two benchmark datasets: MNIST (Deng, 2012) and CIFAR-10 (Krizhevsky et al., 2014). We first detail the experimental setup, and then describe the results and observations stemming from

---

[2]That is, for some $L \geq 0$, by denoting $\mathcal{L}$ the loss function, we have $\|\nabla\mathcal{L}(\mathbf{x}) - \nabla\mathcal{L}(\mathbf{y})\| \leq L \|\mathbf{x} - \mathbf{y}\|$ for all $\mathbf{x}, \mathbf{y} \in \mathbb{R}^d$.

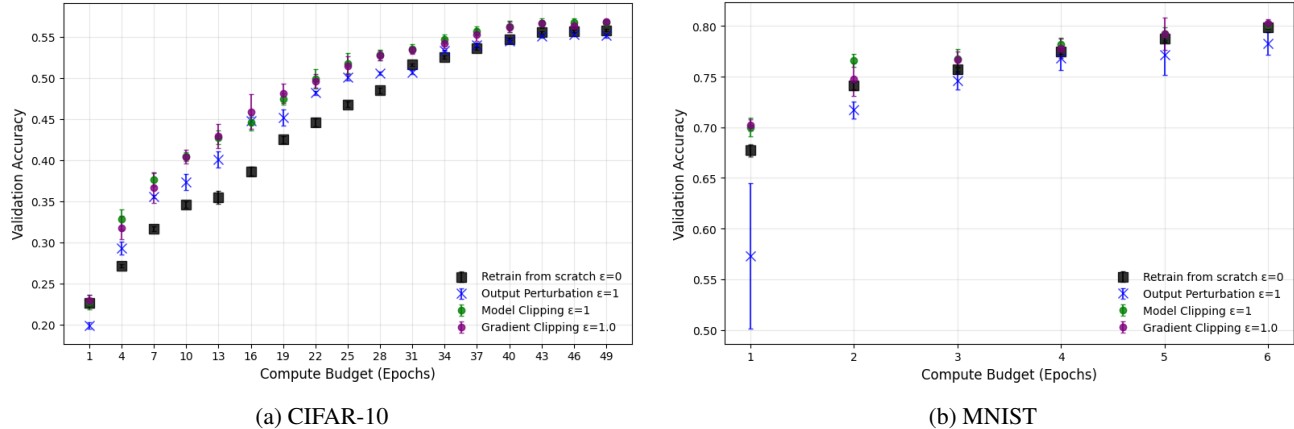

(a) CIFAR-10                                          (b) MNIST

*Figure 1.* Accuracy of *Gradient and Model Clipping* versus compute budget (epochs) on CIFAR-10 (left) and MNIST (right), to satisfy $(1, 10^{-5})$-unlearning. We compare to two baselines: retraining from scratch and output perturbation, detailed in Section 2. Across all the compute budgets gradient and model clipping achieves higher accuracy than the baselines, with the difference being larger for smaller compute budgets.

| Accuracy | **Baselines** | | **Noisy Fine-Tuning (ours)** | |
|---|---|---|---|---|
| | Retrain | Output Perturbation | Gradient Clipping | Model Clipping |
| 30% | 6 | 5 ($\approx$ 16 % faster) | 4 ($\approx$ 33 % faster) | 3 ($\approx$ 50 % faster) |
| 35% | 11 | 7 ($\approx$ 41 % faster) | 6 ($\approx$ 50 % faster) | 6 ($\approx$ 50 % faster) |
| 40% | 18 | 12 ($\approx$ 33 % faster) | 10 ($\approx$ 44 % faster) | 10 ($\approx$ 44 % faster) |
| 45% | 23 | 17 ($\approx$ 26 % faster) | 16 ($\approx$ 30 % faster) | 17 ($\approx$ 26 % faster) |
| 50% | 30 | 25 ($\approx$ 16 % faster) | 23 ($\approx$ 23 % faster) | 24 ($\approx$ 20 % faster) |

*Table 2.* Number of epochs required to reach the target accuracy for our algorithms and the baselines, and their saving compared to retraining from scratch for the CIFAR-10 dataset. *Gradient Clipping* and *Model Clipping* consistently save above 20% of compute, sometimes reaching 50% of compute savings. The output perturbation baseline also consistently improves over the retrain from scratch, however is consistently slower than the Gradient and Model clipping algorithms.

Table 2 and Figures 1 and 2.

### 5.1. Setup

For MNIST, we train a small neural network with two layers and approximately 4,000 parameters. For CIFAR-10, we use a slightly larger network with two convolutional blocks followed by a linear layer, totaling 20,000 parameters. In both cases, the forget set consists of a randomly selected 10% subset of the full dataset.

**Baselines.** We compare our methods against two baselines presented in Section 2: retraining from scratch and output perturbation (1). Retraining from scratch involves fully re-training the model after removing the forget set. Output perturbation applies noise directly to the final model parameters to achieve certified unlearning, before fine-tuning the model on the retain data if the compute budget allows. To

the best of our knowledge, no existing method provides certified unlearning guarantees for non-convex tasks without requiring knowledge of the smoothness constant of the loss function.

**Procedures.** When retraining from scratch, the model is reinitialized using the same distribution as in the original training phase. In all experiments, we first train a model on the entire dataset until convergence. We set $\varepsilon = 1, \delta = 10^{-5}$ for all experiments. For our unlearning algorithms, we continue clipping and adding noise until the desired $(\varepsilon, \delta)$-unlearning guarantee is met. In all experiments, the privacy target is reached before exhausting the iteration budget, in less than 100 iterations (see Appendix B for the exact number of unlearning steps to reach target privacy). We therefore continue fine-tuning the model on the retained dataset without additional noise or clipping, using the same hyperparameters as in retraining from scratch. This means that in

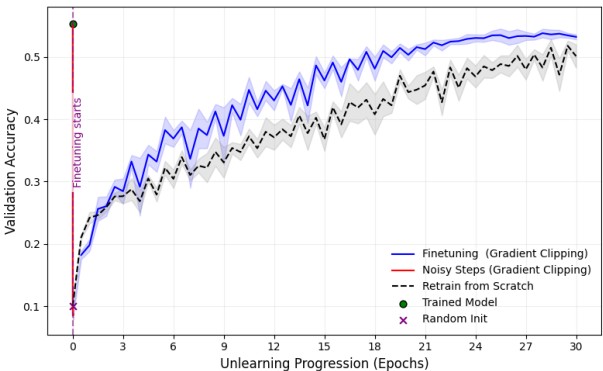

*Figure 2.* Convergence behavior of *Gradient Clipping* with $\gamma = 0.01, C_0 = 20, C_1 = 10, \lambda = 50, \sigma = 0.25$ and the retraining from scratch baseline on the CIFAR-10 dataset. The gradient clipping method is applied for the first 30 iterations, followed by standard fine-tuning. Initially, gradient clipping degrades performance but retains useful information, allowing fine-tuning to recover and surpass the retraining baseline quickly.

all of our experiments unlearning is cheap and effectively finds a new initialization for the finetuning process, that preserves some information from the original model $\hat{x}$. All training, unlearning, and fine-tuning phases use stochastic gradient descent (SGD) with a constant step size. Further experimental details are provided in the appendix.

## 5.2. Results and Observations

We now present our experimental comparison. We compare the algorithms for fixed target accuracy and fixed compute budget, and finally show convergence behavior.

**Fixed target accuracy.** In Table 2, we present the time required for each algorithm to reach the target accuracy for unlearning on CIFAR-10. Our results show that both gradient clipping and model clipping achieve the desired accuracy in a comparable number of steps, significantly outperforming the baseline methods. Notably, compared to retraining from scratch, our algorithms offer substantial computational savings—reducing the required steps by up to 50%. Interestingly, while the simple output perturbation baseline also improves upon retraining from scratch, its efficiency gains are less pronounced. This suggests that while output perturbation approach can be beneficial, more advanced unlearning methods such as gradient and model clipping yield considerably greater improvements.

**Fixed compute budget.** On Figure 1, we show the resulting accuracy for varying compute budgets for gradient and model clipping approaches on MNIST and CIFAR10 datasets. Our experiments demonstrate that our proposed unlearning method, in both its gradient and model clipping

variants, consistently achieves higher accuracy compared to output perturbation and retraining from scratch across all compute budgets. This improvement is particularly pronounced in low-compute settings, where retraining from scratch struggles to recover performance due to the limited number of optimization steps. In contrast, our methods effectively leverage the retained model parameters, enabling faster recovery while ensuring certified unlearning. This provides substantial savings, for example, to reach an accuracy of 40% on CIFAR dataset, both gradient and model clipping needs only 10 epochs, while output perturbation needs 12 epochs (20% longer), and retrain from scratch requires 18 epochs ($\approx 80\%$ longer).

As the compute budget increases, the performance gap between our methods and retraining from scratch gradually narrows. This suggests that while our algorithms provide a strong advantage in resource-constrained scenarios, full retraining may still be the optimal choice given sufficient computing power. However, we note that in practical settings, where compute resources are finite, our approaches offer substantial time savings to reach a particular accuracy.

**Convergence curve.** In Figure 2, we illustrate the convergence behavior for the gradient clipping algorithm on the CIFAR-10 dataset with parameters $\gamma = 0.01, C_0 = 20, C_1 = 10$, and $\lambda = 50$. In that case, unlearning is performed for the first 30 iterations, which significantly decreases the accuracy of the original model to almost zero. However, during the fine-tuning stage, the accuracy quickly catches up and outperforms retraining from scratch in around 1 epoch. This suggests that our stochastic postprocessing approach does not completely erase all prior training. Despite the bad accuracy initially, the model can recover the useful information stored in it quickly. A similar convergence curve is observed in all other settings, as unlearning is always performed for a relatively small number of steps ($< 100$, see Appendix B). These findings highlight the robustness of our approach and its adaptability across different datasets and model architectures.

Overall, we observe that both variants of our method—gradient and model clipping—achieve considerable gains of up to 50% of time savings over the baselines. Further analysis of our results shows that the noise magnitude and clipping strategies play a crucial role in balancing unlearning guarantees with model utility. We found that gradient clipping has a larger range of hyperparameters that achieve an advantage over the baselines, making it easier to tune.

## 5.3. Transfer Learning and Comparison with DP-SGD

To evaluate our methods in more complex settings, we conducted experiments on CIFAR-100 and CIFAR-10 using ResNet architectures (He et al., 2016) pretrained on public

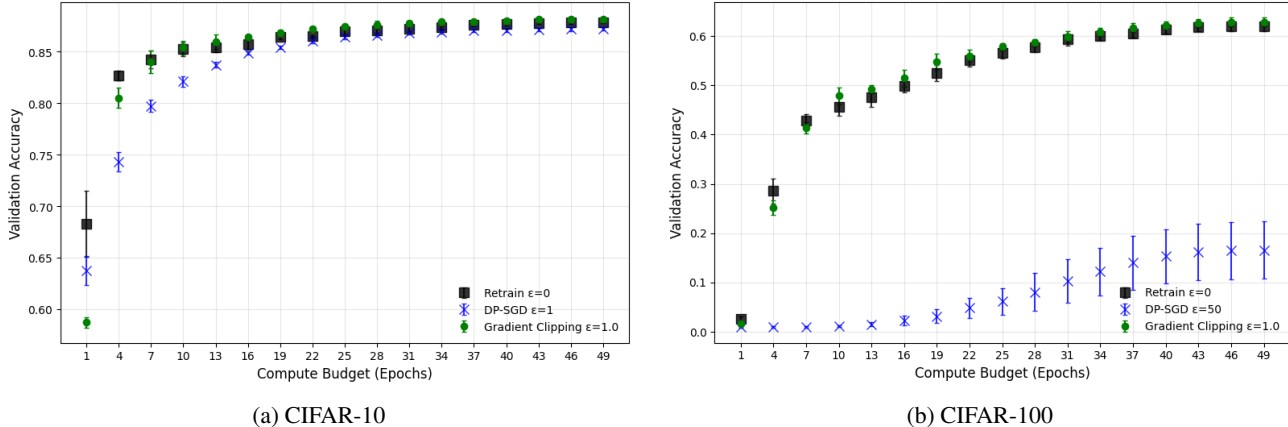

(a) CIFAR-10           (b) CIFAR-100

*Figure 3.* Accuracy of *Gradient Clipping* versus compute budget (epochs) on CIFAR-10 (left) and CIFAR-100 (right) using a ResNet-18 feature extractor pretrained on public data, to satisfy $(1, 10^{-5})$-unlearning.

data (ImageNet (Deng et al., 2009)). This setup, where unlearning is applied to the last few layers of a pretrained model, has become standard in recent certified approximate unlearning works (Guo et al., 2020; Chien et al., 2024), although the latter works focus on a logistic regression task. More precisely, we remove the last layer of ResNet-18 (pretrained on public data) and replace it with a 3-layer fully connected neural network head, which makes the task nonconvex. We first train the head on the full data, and then unlearn the forget data from the head. While we unlearn only the head, we certify the *whole* model because the frozen feature extractor is public and unchanged. In this setting we also compare against DP-SGD with group-privacy baseline as defined in Section 4.2, this produces a certified unlearnt model, so we spend the unlearning compute budget on finetuning the model on the retain data. On CIFAR-10 our method attains 85 % accuracy in 9 epochs, 10 % faster than retrain and 47 % faster than DP-SGD (Fig. 3). The gap widens on CIFAR-100: we reach 60 % accuracy in 32 epochs versus 34 for retrain, while DP-SGD never exceeds 20 % within the 50-epoch budget. The poor DP-SGD curve confirms the theoretical predictions from Sec. 4.2: group-privacy forces $\sqrt{k}$ more noise, $k$ being the number of forget samples, and thus hurts accuracy even under a much weaker privacy budget ($\varepsilon = 50$ vs. our $\varepsilon = 1$)

## 6. Conclusion and Future Work

We introduced a new certified machine unlearning method that provides formal guarantees while remaining broadly applicable to modern neural networks. Our approach leverages the connection between unlearning and privacy amplification through stochastic post-processing, enabling effective removal of data influence without imposing assumptions on the loss function. By applying noisy fine-tuning to the retain

set, our methods achieve both theoretical soundness and practical effectiveness, outperforming existing baselines in empirical evaluations.

Despite these strengths, our approach has certain limitations. First, the effectiveness of our method is constrained by the curse of dimensionality inherent in differential privacy, which can make scaling to models with very large numbers of parameters more challenging. Second, our unlearning framework is designed specifically for stochastic gradient descent (SGD) during the unlearning stage, as we do not retain memory from earlier steps. However, this restriction does not apply to the initial model training or post-unlearning fine-tuning, allowing for flexibility in those phases. Future work could explore extensions to more complex architectures and alternative optimization methods, potentially improving scalability while maintaining strong unlearning guarantees. Our findings highlight the feasibility of certified unlearning in realistic deep learning settings for the first time, offering a promising direction for privacy-preserving and efficient machine learning.

## Impact Statement

This paper presents work whose goal is to advance the field of Machine Learning. There are many potential societal consequences of our work, none of which we feel must be specifically highlighted here.

## Acknowledgements

SK acknowledges support by NSF 2046795 and 2205329, IES R305C240046, ARPA-H, the MacArthur Foundation, Schmidt Sciences, OpenAI, and Stanford HAI. YA acknowledges support by SNSF grant 200021_200477.

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

# A. Proofs

## A.1. Theorem 4.2: Model Clipping

**Preliminaries.** We first recall the hockey-stick divergence and previous results on privacy amplification by stochastic post-processing (Balle et al., 2019).

**Definition A.1** (Hockey-stick divergence). Let $\varepsilon \geq 0$, and $\mu, \nu$ two probability measures defined over $\mathbb{R}^d$. We define

$$\mathrm{E}_\varepsilon(\mu \parallel \nu) := \int_{\mathbb{R}^d} d[\mu - e^\varepsilon \nu]_+ = \sup_{A \subset \mathbb{R}^d} \left( \mu(A) - e^\varepsilon \nu(A) \right),$$

where $[\,\cdot\,]_+ := \max\{0, \cdot\}$.

**Lemma A.2** ((Balle et al., 2019), Theorem 1 (adapted)). *Let $\varepsilon \geq 0$, $K$ be a Markov kernel taking inputs in $\mathbb{R}^d$, and $\mu, \nu$ be two probability distributions over $\mathbb{R}^d$. We have*

$$\mathrm{E}_\varepsilon(\mu K \parallel \nu K) \leq \mathrm{E}_\varepsilon(\mu \parallel \nu) \cdot \sup_{\mathbf{x}_1, \mathbf{x}_2 \in \mathbb{R}^d} \mathrm{E}_\varepsilon(K(\mathbf{x}_1) \parallel K(\mathbf{x}_2)).$$

**Lemma A.3** ((Asoodeh et al., 2020), Lemma 2). *Let $\varepsilon \geq 0$, $\boldsymbol{\mu}_1 \neq \boldsymbol{\mu}_2 \in \mathbb{R}^d$ and $\sigma > 0$. We have*

$$\mathrm{E}_\varepsilon(\mathcal{N}(\boldsymbol{\mu}_1, \sigma^2 \mathbf{I}_d) \parallel \mathcal{N}(\boldsymbol{\mu}_2, \sigma^2 \mathbf{I}_d)) = Q\left( \frac{\varepsilon \sigma}{\|\boldsymbol{\mu}_1 - \boldsymbol{\mu}_2\|} - \frac{\|\boldsymbol{\mu}_1 - \boldsymbol{\mu}_2\|}{2\sigma} \right) - e^\varepsilon Q\left( \frac{\varepsilon \sigma}{\|\boldsymbol{\mu}_1 - \boldsymbol{\mu}_2\|} + \frac{\|\boldsymbol{\mu}_1 - \boldsymbol{\mu}_2\|}{2\sigma} \right),$$

*where for all $t \in \mathbb{R}$, $Q(t) := \frac{1}{\sqrt{\pi}} \int_t^\infty e^{-u^2/2} du$.*

**Lemma A.4** ((Dwork & Roth, 2014), Theorem A.1 (paraphrased)). *Let $\varepsilon \in (0, 1)$ and $\boldsymbol{\mu}_1 \neq \boldsymbol{\mu}_2 \in \mathbb{R}^d$, $\sigma > 0$. We have*

$$\mathrm{E}_\varepsilon(\mathcal{N}(\boldsymbol{\mu}_1, \sigma^2 \mathbf{I}_d) \parallel \mathcal{N}(\boldsymbol{\mu}_2, \sigma^2 \mathbf{I}_d)) \leq 1.25 \exp\left( -\frac{\sigma^2 \varepsilon^2}{2 \|\boldsymbol{\mu}_1 - \boldsymbol{\mu}_2\|^2} \right).$$

**Main proof.** We now proceed to proving the main theorem.

**Theorem 4.2** (Model clipping). *Let $T \geq 1, C_0, C_2, \sigma_0, \varepsilon > 0$, and $\delta \in (0, 1)$. Denote for every $r > 0$,*

$$\theta_\varepsilon(r) := Q\left( \frac{\varepsilon}{r} - \frac{r}{2} \right) - e^\varepsilon Q\left( \frac{\varepsilon}{r} + \frac{r}{2} \right), \tag{9}$$

*where for all $t \in \mathbb{R}$, $Q(t) := \frac{1}{\sqrt{2\pi}} \int_t^\infty e^{-u^2/2} du$.*

*Consider $T$ iterations of the unlearning algorithm defined in (4). We obtain $(\varepsilon, \delta)$-unlearning if*

$$T \geq \frac{\log(1/\delta) + \log \theta_\varepsilon(\frac{2C_0}{\sigma_0})}{\log\left(1/\theta_\varepsilon(\frac{2C_2}{\sigma})\right)}. \tag{10}$$

*In particular, for any $T \geq 1, \varepsilon \in (0, 1)$, it suffices to have*

$$\sigma^2 = \frac{8C_2^2 \ln(1.25)}{\varepsilon^2} \left[ 1 + \frac{1}{T} \left( \ln(1.25/\delta) - \frac{\sigma_0^2 \varepsilon^2}{8C_0^2} \right) \right]. \tag{11}$$

*Proof.* Let $T \geq 1, C_0, C_2, \sigma_0, \gamma > 0$, and $\delta \in (0, 1), \varepsilon \in (0, 3\log(1/\delta))$. Consider $T$ iterations of the unlearning algorithm defined in (4), and analogously define the following sequence initialized at the projected model trained without the forget data $\mathbf{x}_0' := \Pi_{C_0}(\mathcal{A}(\mathcal{D} \setminus \mathcal{D}_f)) + \xi_0, \xi_0 \sim \mathcal{N}(0, \sigma_0^2 \mathbf{I}_d)$:

$$\mathbf{x}_{t+1}' = \Pi_{C_2}(\mathbf{x}_t' - \gamma G(\mathbf{x}_t')) + \xi_t', \qquad \xi_t' \sim \mathcal{N}(0, \sigma^2 \mathbf{I}_d). \tag{15}$$

Recall from Definition A.1 the definition of the hockey-stick divergence $\mathrm{E}_\varepsilon$. Also, we recall from Lemma A.2 that for any Markov kernel $K$:

$$\mathrm{E}_\varepsilon(\mu K \parallel \nu K) \leq \sup_{\mathbf{x}_1, \mathbf{x}_2 \in \mathbb{R}^d} \mathrm{E}_\varepsilon(K(\mathbf{x}_1) \parallel K(\mathbf{x}_2)) \cdot \mathrm{E}_\varepsilon(\mu \parallel \nu).$$

In particular, by introducing $\alpha := \sup_{\mathbf{x}_1, \mathbf{x}_2 \in \mathbb{R}^d} \mathrm{E}_\varepsilon(\mathcal{N}(\Pi_{C_2}(\mathbf{x}_1 - \gamma G(\mathbf{x}_1)), \sigma^2 \mathbf{I}_d) \| \mathcal{N}(\Pi_{C_2}(\mathbf{x}_2 - \gamma G(\mathbf{x}_2)), \sigma^2 \mathbf{I}_d))$ we have

$$\mathrm{E}_\varepsilon(\mathbf{x}_{t+1} \| \mathbf{x}'_{t+1}) \leq \alpha \cdot \mathrm{E}_\varepsilon(\mathbf{x}_t \| \mathbf{x}'_t).$$

Applying the above recursively over $T$ iterations, and denoting $\beta := \mathrm{E}_\varepsilon(\mathcal{N}(\Pi_{C_0}(\mathcal{A}(\mathcal{D})), \sigma_0^2 \mathbf{I}_d) \| \mathcal{N}(\Pi_{C_0}(\mathcal{A}(\mathcal{D} \setminus \mathcal{D}_f)), \sigma_0^2 \mathbf{I}_d))$, yields:

$$\mathrm{E}_\varepsilon(\mathbf{x}_T \| \mathbf{x}'_T) \leq \alpha^T \cdot \mathrm{E}_\varepsilon(\mathbf{x}_0 \| \mathbf{x}'_0) = \alpha^T \cdot \beta.$$

Therefore, in order to satisfy $(\varepsilon, \delta)$-unlearning, it suffices to achieve $\mathrm{E}_\varepsilon(\mathbf{x}_T \| \mathbf{x}'_T) \leq \delta$, which can be achieved by having:

$$T \geq \frac{\log(1/\delta) + \log \beta}{\log(1/\alpha)}.$$

Now, since for any $\mathbf{x}_1, \mathbf{x}_2$ it holds that $\|\Pi_{C_2}(\mathbf{x}_1 - \gamma G(\mathbf{x}_1)) - \Pi_{C_2}(\mathbf{x}_2 - \gamma G(\mathbf{x}_2))\| \leq 2C_2$ and $r \mapsto Q\left(\frac{\varepsilon\sigma}{r} - \frac{r}{2\sigma}\right) - e^\varepsilon Q\left(\frac{\varepsilon\sigma}{r} + \frac{r}{2\sigma}\right)$ is increasing (Asoodeh et al., 2020), using the exact expression of the hockey-stick divergence between Gaussians from Lemma A.3 yields

$$\alpha = \sup_{\mathbf{x}_1, \mathbf{x}_2 \in \mathbb{R}^d} \mathrm{E}_\varepsilon(\mathcal{N}(\Pi_{C_2}(\mathbf{x}_1 - \gamma G(\mathbf{x}_1)), \sigma^2 \mathbf{I}_d) \| \mathcal{N}(\Pi_{C_2}(\mathbf{x}_2 - \gamma G(\mathbf{x}_2)), \sigma^2 \mathbf{I}_d))$$

$$\leq Q\left(\frac{\varepsilon\sigma}{2C_2} - \frac{C_2}{\sigma}\right) - e^\varepsilon Q\left(\frac{\varepsilon\sigma}{2C_2} + \frac{C_2}{\sigma}\right).$$

Similarly, since $\|\Pi_{C_0}(\mathcal{A}(\mathcal{D})) - \Pi_{C_0}(\mathcal{A}(\mathcal{D} \setminus \mathcal{D}_f))\| \leq 2C_0$, we have

$$\beta = \mathrm{E}_\varepsilon(\mathcal{N}(\Pi_{C_0}(\mathcal{A}(\mathcal{D})), \sigma_0^2 \mathbf{I}_d) \| \mathcal{N}(\Pi_{C_0}(\mathcal{A}(\mathcal{D} \setminus \mathcal{D}_f)), \sigma_0^2 \mathbf{I}_d)) \leq Q\left(\frac{\varepsilon\sigma_0}{2C_0} - \frac{C_0}{\sigma_0}\right) - e^\varepsilon Q\left(\frac{\varepsilon\sigma_0}{2C_0} + \frac{C_0}{\sigma_0}\right).$$

Therefore, to achieve $(\varepsilon, \delta)$-unlearning, it suffices to have

$$T \geq \frac{\log(1/\delta) + \log\left(Q\left(\frac{\varepsilon\sigma_0}{2C_0} - \frac{C_0}{\sigma_0}\right) - e^\varepsilon Q\left(\frac{\varepsilon\sigma_0}{2C_0} + \frac{C_0}{\sigma_0}\right)\right)}{-\log\left(Q\left(\frac{\varepsilon\sigma}{2C_2} - \frac{C_2}{\sigma}\right) - e^\varepsilon Q\left(\frac{\varepsilon\sigma}{2C_2} + \frac{C_2}{\sigma}\right)\right)}.$$

Alternatively, using the simpler upper bound from Lemma A.4 on the hockey-stick divergence between Gaussians, we obtain

$$\alpha = \sup_{\mathbf{x}_1, \mathbf{x}_2 \in \mathbb{R}^d} \mathrm{E}_\varepsilon(\mathcal{N}(\Pi_{C_2}(\mathbf{x}_1 - \gamma G(\mathbf{x}_1)), \sigma^2 \mathbf{I}_d) \| \mathcal{N}(\Pi_{C_2}(\mathbf{x}_2 - \gamma G(\mathbf{x}_2)), \sigma^2 \mathbf{I}_d)) \leq 1.25 \exp\left(-\frac{\sigma^2 \varepsilon^2}{8C_2^2}\right).$$

Similarly, we have

$$\beta = \mathrm{E}_\varepsilon(\mathcal{N}(\Pi_{C_0}(\mathcal{A}(\mathcal{D})), \sigma_0^2 \mathbf{I}_d) \| \mathcal{N}(\Pi_{C_0}(\mathcal{A}(\mathcal{D} \setminus \mathcal{D}_f)), \sigma_0^2 \mathbf{I}_d)) \leq 1.25 \exp\left(-\frac{\sigma_0^2 \varepsilon^2}{8C_0^2}\right).$$

Therefore, assuming $\sigma^2 > \frac{8C_2^2 \ln(1.25)}{\varepsilon^2}$, to achieve $(\varepsilon, \delta)$-unlearning, it suffices to have

$$T \geq \frac{\ln(1.25/\delta) - \frac{\sigma_0^2 \varepsilon^2}{8C_0^2}}{\frac{\sigma^2 \varepsilon^2}{8C_2^2} - \ln(1.25)}.$$

This can be rewritten as

$$\sigma^2 \geq \frac{8C_2^2 \ln(1.25)}{\varepsilon^2} + \frac{8C_2^2 \ln(1.25)}{T\varepsilon^2} \left[\ln(1.25/\delta) - \frac{\sigma_0^2 \varepsilon^2}{8C_0^2}\right].$$

This concludes the proof. $\qquad\square$

## A.2. Theorem 4.1: Gradient Clipping

**Preliminaries.** We first recall some important definitions and state useful lemmas before proceeding to the proof of the main theorem. We first recall the definition of the Rényi divergence, which we will mainly use to prove Theorem 4.1.

**Definition A.5** (Rényi divergence). Let $q > 0, q \neq 1$. The $q$-Rényi divergence between two probability distributions $\mu$ and $\nu$ is defined as

$$D_q(\mu \parallel \nu) := \frac{1}{q-1} \log \mathbb{E}_{X \sim \nu} \left( \frac{\mu(X)}{\nu(X)} \right)^q.$$

We recall the *shifted Rényi divergence* introduced by Feldman et al. (2018). For any $z \geq 0, q \geq 1$, and two distributions $\mu, \nu$ defined on $\mathbb{R}^d$, we define

$$D_q^{(z)}(\mu \parallel \nu) := \inf_{\mu' \,:\, W_\infty(\mu',\mu) \leq z} D_q(\mu' \parallel \nu), \tag{16}$$

where $W_\infty(\cdot, \cdot) := \inf_{\omega \in \Gamma(\cdot,\cdot)} \operatorname{ess\,sup}_{(\mathbf{x},\mathbf{y}) \sim \omega} \|\mathbf{x} - \mathbf{y}\|_2$ is the $\infty$-Wasserstein distance, and $\Gamma(\mu', \mu)$ is the collection of couplings of its arguments, i.e., joint measures whose marginals are $\mu'$ and $\mu$ respectively.

**Lemma A.6** ((Feldman et al., 2018), Lemma 20 (adapted)). *Let $q \geq 1, z, a \geq 0$ and $X, Y$ arbitrary random variables. If $\xi, \xi' \sim \mathcal{N}(0, \sigma^2 \mathbf{I}_d), \sigma > 0$, then*

$$D_q^{(z)}(X + \xi \parallel Y + \xi') \leq D_q^{(z+a)}(X \parallel Y) + \frac{qa^2}{2\sigma^2}.$$

**Lemma A.7.** *Let $q \geq 1, z, \rho, C \geq 0, \psi \colon \mathbb{R}^d \to \mathbb{R}^d$ and $X, Y$ arbitrary random variables.*

*If $\psi$ satisfies $\forall \mathbf{x}, \mathbf{x}' \in \mathbb{R}^d, \|\psi(\mathbf{x}') - \psi(\mathbf{x})\| \leq \rho \|\mathbf{x}' - \mathbf{x}\| + s$, then*

$$D_q^{(\rho z + s)}(\psi(X) \parallel \psi(Y)) \leq D_q^{(z)}(X \parallel Y).$$

*Proof.* For any measure $\mu$, we denote by $\psi_{\#}\mu$ the push-forward measure of $\mu$ by $\psi$. Assume that $\psi$ satisfies $\forall \mathbf{x}, \mathbf{x}' \in \mathbb{R}^d, \|\psi(\mathbf{x}') - \psi(\mathbf{x})\| \leq \rho \|\mathbf{x}' - \mathbf{x}\| + s$. By definition of the $\infty$-Wasserstein distance, it follows immediately that

$$W_\infty(\psi_{\#}\mu, \psi_{\#}\nu) \leq \rho \cdot W_\infty(\mu, \nu) + s. \tag{17}$$

Therefore, by definition (16) of the shifted Rényi divergence and using the data processing inequality for Rényi divergences (Van Erven & Harremos, 2014), we have

$$
\begin{aligned}
D_q^{(\rho z + s)}(\psi(X) \parallel \psi(Y)) &= \inf_{\mu' \,:\, W_\infty(\mu', \psi(X)) \leq \rho z + s} D_q(\mu' \parallel \psi(Y)) \\
&\leq \inf_{X' \,:\, W_\infty(\psi(X'), \psi(X)) \leq \rho z + s} D_q(\psi(X') \parallel \psi(Y)) \\
&\leq \inf_{X' \,:\, W_\infty(X', X) \leq z} D_q(\psi(X') \parallel \psi(Y)) && \text{(Inequality (17))} \\
&\leq \inf_{X' \,:\, W_\infty(X', X) \leq z} D_q(X' \parallel Y) && \text{(Data Processing inequality)} \\
&= D_q^{(z)}(X \parallel Y).
\end{aligned}
$$

This concludes the proof. $\qquad\square$

**Lemma A.8.** *Let $\gamma, \lambda \geq 0, G \colon \mathbb{R}^d \to \mathbb{R}^d$ be an arbitrary function, and $\psi \colon \mathbf{x} \mapsto \mathbf{x} - \gamma(\Pi_C(G(\mathbf{x})) + \lambda \mathbf{x})$. Then $\psi$ satisfies:*

$$\forall \mathbf{x}, \mathbf{x}' \in \mathbb{R}^d, \|\psi(\mathbf{x}') - \psi(\mathbf{x})\| \leq |1 - \lambda\gamma| \|\mathbf{x}' - \mathbf{x}\| + 2\gamma C.$$

*Proof.* We have for any $\mathbf{x}, \mathbf{x}' \in \mathbb{R}^d$ that

$$
\begin{aligned}
\|\psi(\mathbf{x}) - \psi(\mathbf{x}')\| &= \|\mathbf{x} - \gamma(\Pi_C(G(\mathbf{x})) + \lambda \mathbf{x}) - \mathbf{x}' + \gamma(\Pi_C(G(\mathbf{x}')) + \lambda \mathbf{x}')\| \\
&\leq |1 - \lambda\gamma| \|\mathbf{x} - \mathbf{x}'\| + \gamma \|\Pi_C(G(\mathbf{x})) - \Pi_C(G(\mathbf{x}'))\| && \text{(Triangle inequality)} \\
&\leq |1 - \lambda\gamma| \|\mathbf{x} - \mathbf{x}'\| + 2\gamma C. && (\|\Pi_C(G(\mathbf{x}))\| \leq C)
\end{aligned}
$$

$\qquad\square$

**Main proof.** We are interested in the following iterative unlearning procedure (generalizing (3) to regularization and varying stepsizes and noise variances), starting from the projected model trained on the full data $\mathbf{x}_0 := \Pi_{C_0}(\mathcal{A}(\mathcal{D}))$, where for all $t \in \{0, \ldots, T-1\}$:

$$\mathbf{x}_{t+1} = \mathbf{x}_t - \gamma_t \left( \Pi_{C_1}(G(\mathbf{x}_t)) + \lambda \mathbf{x}_t \right) + \xi_t, \qquad \xi_t \sim \mathcal{N}(0, \sigma_t^2 \mathbf{I}_d). \tag{18}$$

For the analysis, we analogously define the following sequence initialized at the projected model trained without the forget data $\mathbf{x}_0' := \Pi_{C_0}(\mathcal{A}(\mathcal{D} \setminus \mathcal{D}_f))$:

$$\mathbf{x}_{t+1}' = \mathbf{x}_t' - \gamma_t \left( \Pi_{C_1}(G(\mathbf{x}_t')) + \lambda \mathbf{x}_t' \right) + \xi_t', \qquad \xi_t' \sim \mathcal{N}(0, \sigma_t^2 \mathbf{I}_d). \tag{19}$$

**Theorem A.9.** *Let $T, q \geq 1, \gamma_0, \ldots, \gamma_{T-1} \geq 0, \sigma_0, \ldots, \sigma_{T-1} > 0, \lambda \geq 0$ and consider the two sequences $\{\mathbf{x}_t\}_{0 \leq t \leq T}, \{\mathbf{x}_t'\}_{0 \leq t \leq T}$ as defined above. Denote by $\mathrm{D}_q$ the Rényi divergence of order $q$. Assume that for every $t \in \{0, \ldots, T-1\}, \gamma_t \lambda < 1$. Denote for every $t \in \{0, \ldots, T-1\}, s_t := 2\gamma_t C_1, \rho_t := 1 - \gamma_t \lambda$.*

*If $a_0, \ldots, a_{T-1} \geq 0$ satisfy $\sum_{t=0}^{T-1} \left( \prod_{k=1}^{T-1-t} \rho_k \right) a_t = \left( \prod_{t=0}^{T-1} \rho_t \right) 2C_0 + \sum_{t=0}^{T-1} \left( \prod_{k=1}^{T-1-t} \rho_k \right) s_t$, then*

$$\mathrm{D}_q(\mathbf{x}_T \,\|\, \mathbf{x}_T') \leq \sum_{t=0}^{T-1} \frac{q a_t^2}{2\sigma_t^2}. \tag{20}$$

*In particular, we have*

$$\mathrm{D}_q(\mathbf{x}_T \,\|\, \mathbf{x}_T') \leq \frac{q}{2} \frac{\left[ \left( \prod_{t=0}^{T-1} \rho_t \right) 2C_0 + \sum_{t=0}^{T-1} \left( \prod_{k=1}^{T-1-t} \rho_k \right) s_t \right]^2}{\sum_{t=0}^{T-1} \left( \prod_{k=1}^{T-1-t} \rho_k^2 \right) \sigma_t^2}. \tag{21}$$

*Proof.* Let $t \in \{0, \ldots, T-1\}$. Recall the sequence of iterates defined in (18), and analogously define the following sequence initialized at the projected model trained without the forget data $\mathbf{x}_0' := \Pi_{C_0}(\mathcal{A}(\mathcal{D} \setminus \mathcal{D}_f))$:

$$\mathbf{x}_{t+1}' = \mathbf{x}_t' - \gamma_t \left( \Pi_{C_1}(G(\mathbf{x}_t')) + \lambda \mathbf{x}_t' \right) + \xi_t', \qquad \xi_t' \sim \mathcal{N}(0, \sigma_t^2 \mathbf{I}_d). \tag{22}$$

Therefore, for any $a_t \geq 0$, using the bound above with Lemma A.6 yields

$$\mathrm{D}_q^{(z_{t+1})}\left( \mathbf{x}_{t+1} \,\|\, \mathbf{x}_{t+1}' \right) = \mathrm{D}_q^{(z_{t+1})}(\mathbf{x}_t - \gamma_t \left( \Pi_{C_1}(G(\mathbf{x}_t)) + \lambda \mathbf{x}_t \right) + \xi_t \,\|\, \mathbf{x}_t' - \gamma_t \left( \Pi_{C_1}(G(\mathbf{x}_t')) + \lambda \mathbf{x}_t' \right) + \xi_t')$$

$$\leq \mathrm{D}_q^{(z_{t+1}+a_t)}(\mathbf{x}_t - \gamma_t \left( \Pi_{C_1}(G(\mathbf{x}_t)) + \lambda \mathbf{x}_t \right) \,\|\, \mathbf{x}_t' - \gamma_t \left( \Pi_{C_1}(G(\mathbf{x}_t')) + \lambda \mathbf{x}_t' \right)) + \frac{q a_t^2}{2\sigma_t^2}.$$

Now, using Lemma A.8, and the fact that $\gamma_t < \frac{1}{\lambda}$, we establish that $\psi_t : \mathbf{x} \mapsto \mathbf{x} - \gamma_t \Pi_{C_1}(G(\mathbf{x}))$ satisfies $\forall \mathbf{x}, \mathbf{x}' \in \mathbb{R}^d, \|\psi_t(\mathbf{x}') - \psi_t(\mathbf{x})\| \leq (1 - \lambda \gamma_t) \|\mathbf{x}' - \mathbf{x}\| + 2\gamma_t C_1$. Consequently, denoting $s_t := 2\gamma_t C_1$ and $\rho_t := 1 - \lambda \gamma_t$, using the previous fact and Lemma A.7 in the bound above yields

$$\mathrm{D}_q^{(z_{t+1})}\left( \mathbf{x}_{t+1} \,\|\, \mathbf{x}_{t+1}' \right) \leq \mathrm{D}_q^{(z_{t+1}+a_t)}(\mathbf{x}_t - \gamma_t \left( \Pi_{C_1}(G(\mathbf{x}_t)) + \lambda \mathbf{x}_t \right) \,\|\, \mathbf{x}_t' - \gamma_t \left( \Pi_{C_1}(G(\mathbf{x}_t')) + \lambda \mathbf{x}_t' \right)) + \frac{q a_t^2}{2\sigma_t^2}$$

$$\leq \mathrm{D}_q^{\left( \frac{1}{\rho_t}(z_{t+1}+a_t-s_t) \right)}(\mathbf{x}_t \,\|\, \mathbf{x}_t') + \frac{q a_t^2}{2\sigma_t^2}.$$

By denoting $z_t := \frac{1}{\rho_t}\left( z_{t+1} + a_t - s_t \right)$, we have by recursion over $t \in \{0, \ldots, T-1\}$ for any $z_0, a_0, \ldots, a_T \geq 0$:

$$\mathrm{D}_q^{(z_T)}(\mathbf{x}_T \,\|\, \mathbf{x}_T') \leq \mathrm{D}_q^{(z_0)}(\mathbf{x}_0 \,\|\, \mathbf{x}_0') + \sum_{t=0}^{T-1} \frac{q a_t^2}{2\sigma_t^2}, \tag{23}$$

$$z_T = \left( \prod_{t=0}^{T-1} \rho_t \right) z_0 - \sum_{t=0}^{T-1} \left( \prod_{k=1}^{T-1-t} \rho_k \right) (a_t - s_t). \tag{24}$$

Observe that, upon taking $z_0 = 2C_0$, since $\|\mathbf{x}_0, \mathbf{x}_0'\| \leq 2C_0$, it is immediate from definition (16) that $\mathrm{D}_q^{(z_0)}(\mathbf{x}_0 \| \mathbf{x}_0') = 0$. Additionally, taking $z_T = 0$ in the last equation implies that for all $a_0, \ldots, a_{T-1} \geq 0$ such that

$$\sum_{t=0}^{T-1} \left( \prod_{k=1}^{T-1-t} \rho_k \right) a_t = \left( \prod_{t=0}^{T-1} \rho_t \right) 2C_0 + \sum_{t=0}^{T-1} \left( \prod_{k=1}^{T-1-t} \rho_k \right) s_t, \tag{25}$$

we have

$$\mathrm{D}_q(\mathbf{x}_T \| \mathbf{x}_T') = \mathrm{D}_q^{(0)}(\mathbf{x}_T \| \mathbf{x}_T') \leq \sum_{t=0}^{T-1} \frac{q a_t^2}{2\sigma_t^2}. \tag{26}$$

This concludes the first part of the second statement of the theorem. The second part of the second statement is a direct consequence of setting, for all $t \in \{0, \ldots, T-1\}$,

$$a_t = \left[ \left( \prod_{k=0}^{T-1} \rho_k \right) 2C_0 + \sum_{k=0}^{T-1} \left( \prod_{l=1}^{T-1-k} \rho_l \right) s_k \right] \frac{\left( \prod_{k=1}^{T-1-t} \rho_k \right) \sigma_t^2}{\sum_{k=0}^{T-1} \left( \prod_{l=1}^{T-1-k} \rho_l^2 \right) \sigma_t^2}. \tag{27}$$

$\square$

**Theorem 4.1** (Gradient clipping). *Let $T \geq 1, \gamma, \sigma > 0, \lambda \geq 0, \delta \in (0, 1), \varepsilon \in (0, 3\log(1/\delta))$. Consider $T$ iterations of the unlearning algorithm defined in* (3). *We obtain $(\varepsilon, \delta)$-unlearning if:*

1. *Without regularization ($\lambda = 0$):*

$$\sigma^2 = \frac{9 \log(1/\delta)}{\varepsilon^2 T} \left( C_0 + C_1 \gamma T \right)^2. \tag{5}$$

2. *With regularization ($\lambda > 0$): if $\gamma\lambda \in (\frac{1}{2}, 1)$ and*

$$\sigma^2 = \frac{72 \gamma\lambda \log(1/\delta)}{\varepsilon^2} \left( C_0 \left( 1 - \gamma\lambda \right)^T + \frac{C_1}{\lambda} \right)^2. \tag{6}$$

*Proof.* The proof of the first claim follows immediately by taking constant noise variance, stepsize, and zero regularization in the second statement of Theorem A.9, before converting from Rényi to $(\varepsilon, \delta)$-unlearning using standard conversion methods (Mironov, 2017).

Similarly, the proof of the second claim follows immediately by taking constant noise variance, and stepsize in the second statement of Theorem A.9 (which assumes that $\gamma\lambda < 1$), before converting from Rényi to $(\varepsilon, \delta)$-unlearning using standard conversion methods (Mironov, 2017). Indeed, we then get that it is sufficient to set

$$\sigma^2 \geq \frac{\gamma\lambda(2 - \gamma\lambda)}{2\varepsilon \left( 1 - (1 - \gamma\lambda)^{2T} \right)} \left[ 2C_0 \left( 1 - \gamma\lambda \right)^T + \frac{2C_1}{\lambda} \left( 1 - (1 - \gamma\lambda)^T \right) \right]^2.$$

The right-hand side above can be upper bounded by $\frac{72\gamma\lambda \log(1/\delta)}{\varepsilon^2} \left( C_0 \left( 1 - \gamma\lambda \right)^T + \frac{C_1}{\lambda} \right)^2$ when assuming that $\gamma\lambda \geq \frac{1}{2}$. This concludes the proof. $\square$

# B. Experiments

We use small custom networks for training on MNIST and CIFAR10

```
1  # used for mnist
2  class TinyNet(nn.Module):
3      num_classes: int
4
5      @nn.compact
6      def __call__(self, x, train: bool = True, mutable=None):
7          x = x.reshape((x.shape[0], -1))
8          x = nn.Dense(features=5)(x)
9          x = nn.relu(x)
10         x = nn.Dense(features=self.num_classes)(x)
11         return x
12
13 class CIFAR10TinykNet(nn.Module):
14     num_classes: int
15
16     @nn.compact
17     def __call__(self, x, train: bool = True):
18         he_init = nn.initializers.he_normal()
19         x = nn.Conv(features=32, kernel_size=(3, 3), padding="same", kernel_init=he_init)(
    x)
20         x = nn.relu(x)
21         x = nn.avg_pool(x, window_shape=(2, 2), strides=(2, 2))
22         x = nn.Conv(features=64, kernel_size=(3, 3), padding="same", kernel_init=he_init)(
    x)
23         x = nn.relu(x)
24         x = nn.avg_pool(x, window_shape=(2, 2), strides=(2, 2))
25         x = x.mean(axis=(1, 2))
26         x = nn.Dense(self.num_classes, kernel_init=he_init)(x)
27         return x
```

In Tables 3 and 4 we give complete experimental details for the CIFAR and MNIST experiments.

| Dataset | CIFAR-10 |
|---|---|
| Architecture | Tiny Convolution Net (20k params) |
| Training objective | Cross entropy loss |
| Evaluation objective | Top-1 accuracy |
| Batch size | 128 |
| Training learning rate | 0.1 |
| Training learning rate schedule | Linear One Cycle (Smith & Topin, 2017) |
| Train weight decay | 0.0005 |
| Number of train epochs | 100 |
| Forget set size | 10% |
| Number of unlearning epochs | 50 |
| Noise schedule | constant |
| Unlearning learning rate schedule | constant |
| Post Unlearning learning rate | 0.06 |
| Post Unlearning learning rate schedule | Linear One Cycle |
| Post Unlearning weight decay | 0.0005 |

*Table 3.* Experimental Setting CIFAR10

| Dataset | MNIST |
|---|---|
| Architecture | Tiny 2 Layer Net (4k params) |
| Training objective | Cross entropy loss |
| Evaluation objective | Top-1 accuracy |
| Batch size | 128 |
| Training learning rate | 0.06 |
| Training learning rate schedule | Linear One Cycle (Smith & Topin, 2017) |
| Train weight decay | 0.0005 |
| Number of train epochs | 30 |
| Forget set size | 10% |
| Number of unlearning epochs | 10 |
| Noise schedule | constant |
| Unlearning learning rate schedule | constant |
| Post Unlearning learning rate | 0.06 |
| Post Unlearning learning rate schedule | Linear One Cycle |
| Post Unlearning weight decay | 0.0005 |

*Table 4.* Experimental Setting MNIST

| $\epsilon$ | Compute Budget | $\lambda$ | $C_1$ | $\gamma_t$ | $C_0$ | Unlearning Steps | $\sigma$ |
|---|---|---|---|---|---|---|---|
| 1 | 1 | 10.0 | 100.0 | 0.0001 | 0.01 | 1 | 0.028270 |
| 1 | 2 | 750.0 | 10.0 | 0.0001 | 0.01 | 6 | 0.007752 |
| 1 | 3 | 750.0 | 10.0 | 0.0001 | 0.01 | 6 | 0.007752 |
| 1 | 4 | 750.0 | 10.0 | 0.0001 | 0.01 | 6 | 0.007752 |
| 1 | 5 | 10.0 | 100.0 | 0.0001 | 0.01 | 1 | 0.028270 |
| 1 | 6 | 750.0 | 10.0 | 0.0001 | 0.01 | 6 | 0.007752 |
| 1 | 7 | 10.0 | 100.0 | 0.0001 | 0.01 | 1 | 0.028270 |
| 1 | 8 | 10.0 | 100.0 | 0.0001 | 0.01 | 1 | 0.028270 |
| 1 | 9 | 10.0 | 100.0 | 0.0001 | 0.01 | 1 | 0.028270 |
| 1 | 10 | 10.0 | 100.0 | 0.0001 | 0.01 | 1 | 0.028270 |

*Table 5.* Hyperparameters for Gradient Clipping MNIST

| $\epsilon$ | Compute Budget | $C_2$ | $\sigma$ | $\eta$ | $\lambda$ | Unlearning Steps |
|---|---|---|---|---|---|---|
| 1 | 1 | 0.001 | 0.01 | 0.0001 | 900.00 | 1 |
| 1 | 2 | 0.001 | 0.01 | 0.0001 | 900.00 | 1 |
| 1 | 3 | 0.001 | 0.01 | 0.0001 | 900.00 | 1 |
| 1 | 4 | 0.001 | 0.01 | 0.0001 | 500.00 | 1 |
| 1 | 5 | 0.001 | 0.01 | 0.0100 | 0.01 | 1 |
| 1 | 6 | 0.010 | 0.01 | 0.0010 | 900.00 | 6 |
| 1 | 7 | 0.001 | 0.01 | 0.0010 | 10.00 | 1 |
| 1 | 8 | 0.001 | 0.01 | 0.0100 | 900.00 | 1 |
| 1 | 9 | 0.001 | 0.01 | 0.0001 | 10.00 | 1 |
| 1 | 10 | 0.001 | 0.01 | 0.0010 | 10.00 | 1 |

*Table 6.* Hyperparameters for Model Clipping MNIST

| $\epsilon$ | Compute Budget (epochs) | $\lambda$ | $C_1$ | $\gamma$ | $C_0$ | Unlearning Steps | $\sigma$ |
|---|---|---|---|---|---|---|---|
| 1 | 1 | 200.0 | 100.0 | 0.0010 | 0.1 | 1 | 0.254558 |
| 1 | 4 | 50.0 | 10.0 | 0.0100 | 1.0 | 5 | 0.275702 |
| 1 | 7 | 50.0 | 10.0 | 0.0100 | 20.0 | 11 | 0.256790 |
| 1 | 10 | 50.0 | 100.0 | 0.0001 | 0.1 | 10 | 0.088213 |
| 1 | 13 | 1.0 | 10.0 | 0.0010 | 0.1 | 10 | 0.089197 |
| 1 | 16 | 50.0 | 10.0 | 0.0010 | 0.1 | 10 | 0.077256 |
| 1 | 19 | 1.0 | 10.0 | 0.0010 | 0.1 | 10 | 0.089197 |
| 1 | 22 | 50.0 | 100.0 | 0.0001 | 0.1 | 10 | 0.088213 |
| 1 | 25 | 500.0 | 100.0 | 0.0010 | 1.0 | 5 | 0.275702 |
| 1 | 28 | 50.0 | 10.0 | 0.0100 | 20.0 | 11 | 0.256790 |
| 1 | 31 | 500.0 | 100.0 | 0.0010 | 20.0 | 11 | 0.256790 |
| 1 | 34 | 50.0 | 1.0 | 0.0010 | 1.0 | 93 | 0.012501 |
| 1 | 37 | 50.0 | 100.0 | 0.0010 | 0.1 | 1 | 0.275772 |
| 1 | 40 | 500.0 | 100.0 | 0.0010 | 1.0 | 5 | 0.275702 |
| 1 | 43 | 1.0 | 10.0 | 0.0100 | 0.1 | 1 | 0.281429 |
| 1 | 46 | 500.0 | 100.0 | 0.0010 | 20.0 | 11 | 0.256790 |
| 1 | 49 | 1.0 | 10.0 | 0.0100 | 0.1 | 1 | 0.281429 |

*Table 7.* Hyperparameters for Gradient Clipping CIFAR

| $\epsilon$ | Compute Budget (epochs) | $C_2$ | $\sigma$ | $\eta$ | $\lambda$ | Unlearning Steps |
|---|---|---|---|---|---|---|
| 1 | 1 | 0.200 | 0.2 | 0.0100 | 100.0 | 6 |
| 1 | 4 | 0.500 | 0.5 | 0.0010 | 10.0 | 6 |
| 1 | 7 | 0.500 | 0.5 | 0.0010 | 10.0 | 6 |
| 1 | 10 | 0.625 | 0.5 | 0.0001 | 100.0 | 9 |
| 1 | 13 | 0.625 | 0.5 | 0.0010 | 100.0 | 9 |
| 1 | 16 | 0.625 | 0.5 | 0.0001 | 0.0 | 9 |
| 1 | 19 | 0.500 | 0.5 | 0.0001 | 0.0 | 6 |
| 1 | 22 | 0.500 | 0.5 | 0.0010 | 0.0 | 6 |
| 1 | 25 | 0.625 | 0.5 | 0.0001 | 10.0 | 9 |
| 1 | 28 | 0.625 | 0.5 | 0.0010 | 100.0 | 9 |
| 1 | 31 | 0.625 | 0.5 | 0.0001 | 1.0 | 9 |
| 1 | 34 | 0.975 | 0.5 | 0.0001 | 10.0 | 36 |
| 1 | 37 | 0.625 | 0.5 | 0.0100 | 10.0 | 9 |
| 1 | 40 | 0.975 | 0.5 | 0.0001 | 1.0 | 36 |
| 1 | 43 | 0.500 | 0.5 | 0.0001 | 10.0 | 6 |
| 1 | 46 | 0.975 | 0.5 | 0.0001 | 10.0 | 36 |
| 1 | 49 | 0.500 | 0.5 | 0.0100 | 100.0 | 6 |

*Table 8.* Hyperparameters for Model Clipping CIFAR

| $\epsilon$ | Compute Budget (epochs) | $C_0$ | $\sigma$ |
|---|---|---|---|
| 1 | 1 | 1.00 | 9.689610 |
| 1 | 4 | 0.10 | 0.968961 |
| 1 | 7 | 0.10 | 0.968961 |
| 1 | 10 | 0.10 | 0.968961 |
| 1 | 13 | 0.10 | 0.968961 |
| 1 | 16 | 0.10 | 0.968961 |
| 1 | 19 | 0.10 | 0.968961 |
| 1 | 22 | 0.10 | 0.968961 |
| 1 | 25 | 0.10 | 0.968961 |
| 1 | 28 | 0.10 | 0.968961 |
| 1 | 31 | 0.10 | 0.968961 |
| 1 | 34 | 0.10 | 0.968961 |
| 1 | 37 | 0.10 | 0.968961 |
| 1 | 40 | 0.01 | 0.096896 |
| 1 | 43 | 0.01 | 0.096896 |
| 1 | 46 | 0.10 | 0.968961 |
| 1 | 49 | 0.01 | 0.096896 |

*Table 9.* Hyperparameters for Output Perturbation CIFAR

| $\epsilon$ | Compute Budget (epochs) | $C_0$ | $\sigma$ |
|---|---|---|---|
| 1 | 1 | 0.01 | 0.096896 |
| 1 | 2 | 0.01 | 0.096896 |
| 1 | 3 | 0.01 | 0.096896 |
| 1 | 4 | 0.01 | 0.096896 |
| 1 | 5 | 0.01 | 0.096896 |
| 1 | 6 | 0.01 | 0.096896 |
| 1 | 7 | 0.01 | 0.096896 |
| 1 | 8 | 0.01 | 0.096896 |
| 1 | 9 | 0.01 | 0.096896 |
| 1 | 10 | 0.01 | 0.096896 |

*Table 10.* Hyperparameters for Output Perturbation MNIST

## C. Transfer Learning Experiments

We use small three layer network as the head on top of a frozen pretrained (on Imagenet) ResNet18 backbone for transfer learning experiments on CIFAR-10 and CIFAR-100

```
class ThreeLayerNN(nn.Module):
    num_classes: int

    @nn.compact
    def __call__(self, x, train: bool = True, mutable=None):
        x = x.reshape((x.shape[0], -1))
        x = nn.Dense(features=32)(x)
        x = nn.relu(x)
        x = nn.Dense(features=32)(x)
        x = nn.relu(x)
        x = nn.Dense(features=self.num_classes)(x)
        return x
```

In Tables 11, 12, and 13 we give complete experimental details for the CIFAR-10 and CIFAR-100 transfer learning experiments.

| Architecture | Frozen Resnet-18 Backbone + 3 Layer NN |
|---|---|
| Training objective | Cross entropy loss |
| Evaluation objective | Top-1 accuracy |
| Batch size | 128 |
| Training learning rate | 0.1 |
| Training learning rate schedule | Linear One Cycle (Smith & Topin, 2017) |
| Train weight decay | 0.0005 |
| Number of train epochs | 100 |
| DP-SGD $\|.\|_2 -$clip | 0.5 |
| DP-SGD target group $\varepsilon$ | 50 |
| DP-SGD target group $\delta$ | 0.00001 |
| Forget set size | 10% |
| DP-SGD forget set size | 0.5% |
| Number of unlearning epochs | 50 |
| Noise schedule | constant |
| Unlearning learning rate schedule | constant |
| Post Unlearning learning rate | 0.06 |
| Post Unlearning learning rate schedule | Linear One Cycle |
| Post Unlearning weight decay | 0.0005 |

*Table 11.* Experimental Setting CIFAR-10 and CIFAR-100

| $\epsilon$ | Compute Budget (epochs) | $\lambda$ | $C_1$ | $\gamma$ | $C_0$ | Unlearning Steps | $\sigma$ |
|---|---|---|---|---|---|---|---|
| 1 | 1 | 500.0 | 100.0 | 0.001 | 0.01 | 1 | 0.148492 |
| 1 | 4 | 100.0 | 10.0 | 0.0001 | 0.01 | 10 | 0.008698 |
| 1 | 7 | 0.50 | 1.0 | 0.001 | 0.01 | 10 | 0.008932 |
| 1 | 10 | 0.50 | 10.0 | 0.0001 | 0.01 | 10 | 0.008943 |
| 1 | 13 | 0.50 | 10.0 | 0.0001 | 0.01 | 10 | 0.008943 |
| 1 | 16 | 10.0 | 10.0 | 0.001 | 0.01 | 1 | 0.028143 |
| 1 | 19 | 10.0 | 10.0 | 0.001 | 0.01 | 1 | 0.028143 |
| 1 | 22 | 10.0 | 10.0 | 0.001 | 0.01 | 1 | 0.028143 |
| 1 | 25 | 10.0 | 10.0 | 0.001 | 0.01 | 1 | 0.028143 |
| 1 | 28 | 10.0 | 10.0 | 0.001 | 0.01 | 1 | 0.028143 |
| 1 | 31 | 100.0 | 100.0 | 0.0001 | 0.01 | 1 | 0.028143 |
| 1 | 34 | 10.0 | 1.0 | 0.01 | 0.01 | 1 | 0.026870 |
| 1 | 37 | 0.50 | 10.0 | 0.001 | 0.01 | 1 | 0.028277 |
| 1 | 40 | 0.50 | 10.0 | 0.001 | 0.01 | 1 | 0.028277 |
| 1 | 43 | 0.50 | 10.0 | 0.001 | 0.01 | 1 | 0.028277 |
| 1 | 46 | 0.50 | 10.0 | 0.001 | 0.01 | 1 | 0.028277 |
| 1 | 49 | 0.50 | 10.0 | 0.001 | 0.01 | 1 | 0.028277 |

*Table 12.* Hyperparameters for Gradient Clipping Transfer Learning CIFAR-10

| $\epsilon$ | Compute Budget (epochs) | $\lambda$ | $C_1$ | $\gamma$ | $C_0$ | Unlearning Steps | $\sigma$ |
|---|---|---|---|---|---|---|---|
| 1 | 1 | 500 | 100 | 0.001 | 0.01 | 1 | 0.148492 |
| 1 | 4 | 0.50 | 1 | 0.001 | 0.01 | 10 | 0.008932 |
| 1 | 7 | 10 | 1 | 0.001 | 0.01 | 10 | 0.008698 |
| 1 | 10 | 10 | 0.10 | 0.01 | 0.10 | 30 | 0.008524 |
| 1 | 13 | 10 | 10 | 0.0001 | 0.01 | 10 | 0.008920 |
| 1 | 16 | 100 | 1 | 0.001 | 0.10 | 30 | 0.008524 |
| 1 | 19 | 0.50 | 1 | 0.001 | 0.01 | 10 | 0.008932 |
| 1 | 22 | 500 | 10 | 0.0001 | 0.01 | 10 | 0.007726 |
| 1 | 25 | 100 | 1 | 0.001 | 0.10 | 30 | 0.008524 |
| 1 | 28 | 0.50 | 1 | 0.001 | 0.01 | 10 | 0.008932 |
| 1 | 31 | 500 | 10 | 0.001 | 1 | 10 | 0.025667 |
| 1 | 34 | 0.50 | 10 | 0.0001 | 0.01 | 10 | 0.008943 |
| 1 | 37 | 500 | 10 | 0.001 | 1 | 10 | 0.025667 |
| 1 | 40 | 500 | 10 | 0.001 | 1 | 10 | 0.025667 |
| 1 | 43 | 500 | 10 | 0.001 | 1 | 10 | 0.025667 |
| 1 | 46 | 500 | 10 | 0.001 | 1 | 10 | 0.025667 |
| 1 | 49 | 500 | 10 | 0.001 | 1 | 10 | 0.025667 |

*Table 13.* Hyperparameters for Gradient Clipping Transfer Learning CIFAR-100

## D. $\varepsilon$ Sweep

In this section, we evaluate how the choice of $\epsilon$ affects the performance of our algorithm. For that, in addition to $\epsilon = 1$ used in the paper, we plot the performance of the gradient clipping algorithm (3) for $\epsilon = 0.1$ and $\epsilon = 10$. We kept fixed $\delta = 10^{-5}$ for all of the epsilons. See Figure 4 for results. We can see that $\epsilon = 0.1$ degrades the performance of our algorithm significantly compared to $\epsilon = 1$. There is very little difference between $\varepsilon = 1$ and $\varepsilon = 10$ but a performance penalty for $\varepsilon = 0.1$ that is more visible with the harder task (CIFAR-10).

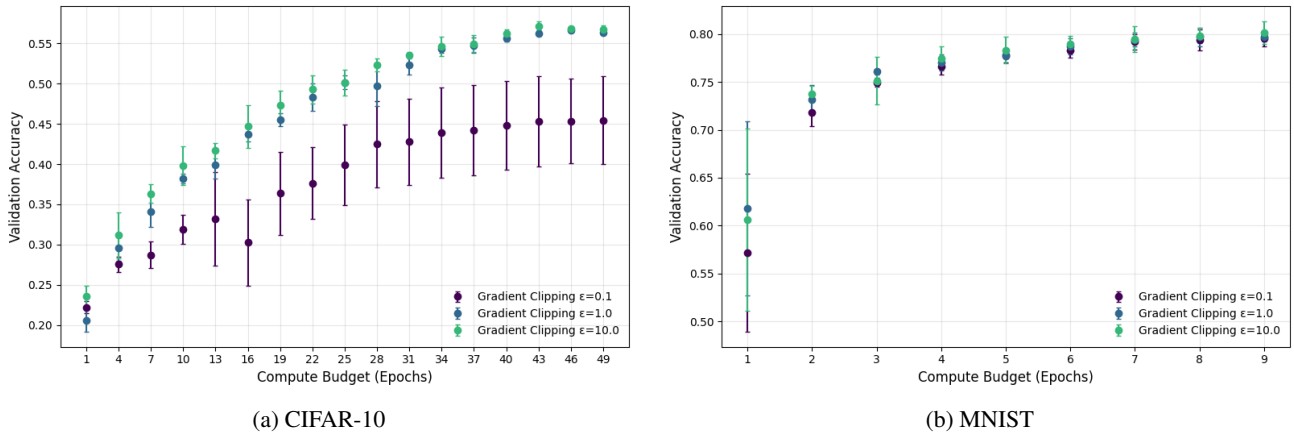

(a) CIFAR-10          (b) MNIST

*Figure 4.* Accuracy of *Gradient Clipping* versus compute budget (epochs) on CIFAR-10 (left) and MNIST (right), to satisfy $(\varepsilon, 10^{-5})$-unlearning for $\varepsilon \in \{0.1, 1, 10\}$.

| Compute Budget (epochs) | $\epsilon$ | $\lambda$ | $C_1$ | $\gamma$ | $C_0$ | Unlearning Steps | $\sigma$ |
|---|---|---|---|---|---|---|---|
| 1 | 0.1 | 500 | 100 | 0.001 | 1 | 5 | 0.871847 |
| 4 | 0.1 | 500 | 100 | 0.001 | 1 | 5 | 0.871847 |
| 7 | 0.1 | 500 | 100 | 0.001 | 1 | 5 | 0.871847 |
| 10 | 0.1 | 500 | 100 | 0.001 | 1 | 5 | 0.871847 |
| 13 | 0.1 | 500 | 100 | 0.001 | 1 | 5 | 0.871847 |
| 16 | 0.1 | 500 | 100 | 0.001 | 1 | 5 | 0.871847 |
| 19 | 0.1 | 500 | 100 | 0.001 | 1 | 5 | 0.871847 |
| 22 | 0.1 | 500 | 100 | 0.001 | 1 | 5 | 0.871847 |
| 25 | 0.1 | 500 | 100 | 0.001 | 1 | 5 | 0.871847 |
| 28 | 0.1 | 500 | 100 | 0.001 | 1 | 5 | 0.871847 |
| 31 | 0.1 | 500 | 100 | 0.001 | 1 | 5 | 0.871847 |
| 34 | 0.1 | 500 | 100 | 0.001 | 1 | 5 | 0.871847 |
| 37 | 0.1 | 500 | 100 | 0.001 | 1 | 5 | 0.871847 |
| 40 | 0.1 | 500 | 100 | 0.001 | 1 | 5 | 0.871847 |
| 43 | 0.1 | 500 | 100 | 0.001 | 1 | 5 | 0.871847 |
| 46 | 0.1 | 500 | 100 | 0.001 | 1 | 5 | 0.871847 |
| 49 | 0.1 | 500 | 100 | 0.001 | 1 | 5 | 0.871847 |
| 1 | 1 | 500 | 100 | 0.001 | 1 | 5 | 0.275702 |
| 4 | 1 | 1 | 10 | 0.001 | 0.1 | 10 | 0.089197 |
| 7 | 1 | 500 | 100 | 0.001 | 1 | 5 | 0.275702 |
| 10 | 1 | 1 | 10 | 0.001 | 0.1 | 10 | 0.089197 |
| 13 | 1 | 500 | 100 | 0.001 | 1 | 5 | 0.275702 |
| 16 | 1 | 500 | 100 | 0.001 | 1 | 5 | 0.275702 |
| 19 | 1 | 500 | 100 | 0.001 | 1 | 5 | 0.275702 |
| 22 | 1 | 500 | 100 | 0.001 | 1 | 5 | 0.275702 |
| 25 | 1 | 500 | 100 | 0.001 | 1 | 5 | 0.275702 |
| 28 | 1 | 500 | 100 | 0.001 | 1 | 5 | 0.275702 |
| 31 | 1 | 500 | 100 | 0.001 | 1 | 5 | 0.275702 |
| 34 | 1 | 500 | 100 | 0.001 | 1 | 5 | 0.275702 |
| 37 | 1 | 500 | 100 | 0.001 | 1 | 5 | 0.275702 |
| 40 | 1 | 500 | 100 | 0.001 | 1 | 5 | 0.275702 |
| 43 | 1 | 500 | 100 | 0.001 | 1 | 5 | 0.275702 |
| 46 | 1 | 500 | 100 | 0.001 | 1 | 5 | 0.275702 |
| 49 | 1 | 500 | 100 | 0.001 | 1 | 5 | 0.275702 |
| 1 | 10 | 1 | 100 | 0.1 | 0.1 | 1 | 4.512385 |
| 4 | 10 | 1 | 10 | 0.1 | 0.1 | 1 | 0.487463 |
| 7 | 10 | 0.1 | 10 | 0.1 | 1 | 1 | 0.889955 |
| 10 | 10 | 0.1 | 10 | 0.1 | 0.1 | 1 | 0.491488 |
| 13 | 10 | 0.1 | 10 | 0.1 | 1 | 1 | 0.889955 |
| 16 | 10 | 0.1 | 10 | 0.1 | 0.1 | 1 | 0.491488 |
| 19 | 10 | 1 | 10 | 0.1 | 0.1 | 1 | 0.487463 |
| 22 | 10 | 0.1 | 10 | 0.1 | 0.1 | 1 | 0.491488 |
| 25 | 10 | 0.1 | 10 | 0.1 | 1 | 1 | 0.889955 |
| 28 | 10 | 0.1 | 0.01 | 0.1 | 0.1 | 70 | 0.007260 |
| 31 | 10 | 1 | 0.1 | 0.1 | 10 | 53 | 0.026744 |
| 34 | 10 | 25 | 5 | 0.01 | 10 | 19 | 0.071419 |
| 37 | 10 | 0.1 | 10 | 0.1 | 0.1 | 1 | 0.491488 |
| 40 | 10 | 1 | 10 | 0.001 | 0.1 | 10 | 0.028207 |
| 43 | 10 | 25 | 5 | 0.01 | 10 | 19 | 0.071419 |
| 46 | 10 | 25 | 5 | 0.01 | 10 | 19 | 0.071419 |
| 49 | 10 | 1 | 0.1 | 0.1 | 1 | 30 | 0.026955 |

*Table 14.* Hyperparameters for Gradient Clipping CIFAR-10 for $\varepsilon \in \{10, 1, 0.1\}$

| $\epsilon$ | Compute Budget (epochs) | $\lambda$ | $C_1$ | $\gamma$ | $C_0$ | Unlearning Steps | $\sigma$ |
|---|---|---|---|---|---|---|---|
| 10 | 1 | 750 | 10 | 0.0001 | 0.01 | 6 | 0.002451 |
| 10 | 2 | 750 | 10 | 0.0001 | 0.01 | 6 | 0.002451 |
| 10 | 3 | 750 | 10 | 0.0001 | 0.01 | 6 | 0.002451 |
| 10 | 4 | 750 | 10 | 0.0001 | 0.01 | 6 | 0.002451 |
| 10 | 5 | 10 | 100 | 0.0001 | 0.01 | 1 | 0.008940 |
| 10 | 6 | 750 | 10 | 0.0001 | 0.01 | 6 | 0.002451 |
| 10 | 7 | 10 | 100 | 0.0001 | 0.01 | 1 | 0.008940 |
| 10 | 8 | 750 | 10 | 0.0001 | 0.01 | 6 | 0.002451 |
| 10 | 9 | 10 | 100 | 0.0001 | 0.01 | 1 | 0.008940 |
| 1 | 1 | 10 | 100 | 0.0001 | 0.01 | 1 | 0.028270 |
| 1 | 2 | 750 | 10 | 0.0001 | 0.01 | 6 | 0.007752 |
| 1 | 3 | 750 | 10 | 0.0001 | 0.01 | 6 | 0.007752 |
| 1 | 4 | 750 | 10 | 0.0001 | 0.01 | 6 | 0.007752 |
| 1 | 5 | 10 | 100 | 0.0001 | 0.01 | 1 | 0.028270 |
| 1 | 6 | 10 | 100 | 0.0001 | 0.01 | 1 | 0.028270 |
| 1 | 7 | 750 | 10 | 0.0001 | 0.01 | 6 | 0.007752 |
| 1 | 8 | 10 | 100 | 0.0001 | 0.01 | 1 | 0.028270 |
| 1 | 9 | 10 | 100 | 0.0001 | 0.01 | 1 | 0.028270 |
| 0.1 | 1 | 10 | 100 | 0.0001 | 0.01 | 1 | 0.089398 |
| 0.1 | 2 | 10 | 100 | 0.0001 | 0.01 | 1 | 0.089398 |
| 0.1 | 3 | 10 | 100 | 0.0001 | 0.01 | 1 | 0.089398 |
| 0.1 | 4 | 750 | 10 | 0.0001 | 0.01 | 6 | 0.024514 |
| 0.1 | 5 | 10 | 100 | 0.0001 | 0.01 | 1 | 0.089398 |
| 0.1 | 6 | 10 | 100 | 0.0001 | 0.01 | 1 | 0.089398 |
| 0.1 | 7 | 10 | 100 | 0.0001 | 0.01 | 1 | 0.089398 |
| 0.1 | 8 | 10 | 100 | 0.0001 | 0.01 | 1 | 0.089398 |
| 0.1 | 9 | 10 | 100 | 0.0001 | 0.01 | 1 | 0.089398 |

*Table 15.* Hyperparameters for Gradient Clipping MNIST for $\varepsilon \in \{10, 1, 0.1\}$

