# OpenReview forum: "Certified Unlearning for Neural Networks"
_ICML.cc/2025/Conference — ICML 2025 poster_

### Official Review · Reviewer_z3Fs · 2025-02-24

**Overall Recommendation:** 4

**Summary:**

This paper proposes to analyze the formal unlearning guarantees of two varieties of clipped noisy finetuning (either model or gradient clipping) by using recent post-processing DP analyses. Specifically they propose to first project and add noise to the original model, and then apply $T$ steps of clipped (either model weights or gradients) noisy SGD on the retain dataset. The analysis then follows from several techniques of privacy amplification from the initial DP guarantee given by the initial projection and Gaussian noise. This is claimed to be the first algorithm agnostic to the original training algorithm. However this is for a definition of unlearning that is different to the unlearning definitions used in past work. Experimental results for the method are presented for CIFAR10 and MNIST, and they observe they improve efficiency over retraining from scratch (though retraining from scratch provides different unlearning guarantees).

## Update after Rebuttal

The rebuttal helped clarify the literature from which the proposed definition came from, and I raised my initial score given this. The authors now also provide experiments comparing to DP-SGD and on more datasets, and I raised my score once more to an accept given this. While I did not follow why there was additional fine-tuning for DP-SGD in this comparison, I trust the authors will explain more in the camera-ready.

I hope the authors will incorporate much of our discussion into the camera-ready, as to also help future readers understand the subtle differences between the various unlearning definitions, and acknowledge potential limitations (which are open-problems). In particular, that this post-processing definition is weaker, but it may still be enough for some settings. I now believe one of the key contributions of this paper is motivating future study on this definition, and perhaps stating this explicitly will help future readers. For example, maybe something along the lines of "We hope future work studies applications for post-processing unlearning, given this paper showed that it allows for more efficient unlearning, with guarantees, and agnostic to the original training algorithm."

**Claims And Evidence:**

The paper claims Definition 2.1 paraphrases the DP inspired unlearning definitions presented in  (Ginart et al 2019) and (Guo et al, 2019). This is incorrect, as in those papers $\bar{A} = A$, i.e., the certifying algorithm must be the original training algorithm, while in this paper they let $\bar{A}$ be free and implicitly use $\bar{A} = U(A(D \setminus D_F), D \setminus D_f, \emptyset)$. Note $\bar{A} = A$ is widely the definition used for unlearning, even in adaptive settings: see “Adaptive Machine Unlearning”  (Gupta et al, 2021). To make their method fit the unlearning definition of past work, we would need the original training algorithm be $A = U(A(D \setminus D_F), D \setminus D_f, \emptyset)$, which in the context of the methods in the paper, requires an additional T steps of projected noisy SGD after the original training run. So their algorithm does require assumptions on training (the authors claim otherwise). However, the impact of this additional assumption on the performance of the models and compute to unlearn is not evaluated.


This discrepancy also means the proposed methods should be compared to past certified unlearning work that also modify the training algorithm to have faster unlearning to justify the improvements of the method: e.g., SISA as proposed by (Bourtoule et al, 2021), or naive DP training. No such comparisons are made.

I point the authors to “On The Necessity of Auditable Algorithmic Definitions for Machine Unlearning” (Thudi et al, 2022) for results motivating why unlearning is widely defined with a fixed training algorithm, and is not just a property of the final model.

**Essential References Not Discussed:**

No essential references missing that I noticed, though as pointed earlier, past work is misrepresented.

**Experimental Designs Or Analyses:**

I have several questions regarding how hyperparameters were selected, and the apparent weak performance of the models on CIFAR10 and MNIST. In particular, the model used only reaches $\sim 55$% accuracy on CIFAR10 and $\sim 80$% on MNIST, while reaching $>90$% is widely standard in the literature for both datasets when using ResNets. Given their unlearning algorithm actually presents changes to the training algorithm, I believe further exploration of the impact to performance of their method is necessary (as we cannot assume it will provide the same performance as current SOTA training algorithms).

**Methods And Evaluation Criteria:**

The evaluation does not capture the fact the method requires modifications to the training algorithm. Specifically:

1) No comparison to other certified approaches which modify the training algorithm are made

2) No analysis/experiments of the impact on the required additional noisy projected fine-tuning is presented

**Other Comments Or Suggestions:**

What follows are potential changes to the paper to remedy issues with the current claims. I am willing to revisit my score if the the issues I've raised regarding the claims are addressed.

1) Rephrase Definition 2.1 to be the same as past work, and state explicitly what training algorithm you are unlearning in the theorem statements. Alternatively, present the definition as a new unlearning definition, however given past negative results on auditing unlearning using just model weights, I currently find this definition hard to justify.

2) Provide experiments exploring the impact of the additional noisy clipped fine-tuning needed to the training algorithm across models and datasets. In particular consider evaluating performance degradation, and account for the additional costs to training this presents compared to naively retraining. I believe it simply doubles the current cost analysis, which would mean the method is less efficient than retraining, but the additional training cost is a one-time cost and may be less important over many unlearning requests.

3) Given the method requires modifications to training, provide a more detailed comparison to the standard (modifying training) exact unlearning algorithm SISA. Note Figure 6 in (Bourtoule et al, 2021) suggests it can cut unlearning costs by $⅓$ with minimal performance degradation for image classification, though as the size of the forget set grows their unlearning cost increases and so eventually the methods in this paper could be more efficient.

**Other Strengths And Weaknesses:**

Strengths:

1) The proof techniques seemed a novel application of past ideas (despite the correction needed)

2) If the claims are corrected as suggested below, I believe it is possible this also improve the state of certifiably unlearning large groups of data with modifications to the training algorithms (where SISA does not scale well)

Weaknesses:

1) The current claims are incorrect and consequently misrepresent the past iterature on unlearning

**Questions For Authors:**

I described my main concerns in the previous sections, but now list specific (but more minor) questions:

1) Why is the accuracy on CIFAR10 and MNIST much lower than standard ResNet results? I understand the architecture is relatively small to the ResNets commonly used, but is there a specific reason not to use ResNet for these experiments? My current hypothesis for the results could be that the method suffers similar performance degradation as DP training.
2) How were the hyperparameters chosen for the figures in the main body? I found the tables of hyperparameters in the appendix but did not connect how the tables were used to choose the final hyperparameters.

**Relation To Broader Scientific Literature:**

Unlearning is proposed as a technique to meet privacy and copyright legislation ( Machine Unlearning Doesn't Do What You Think: Lessons for Generative AI Policy, Research, and Practice” (Cooper et al., 2024) ) to address and detect data poisoning (“Threats,
attacks, and defenses in machine unlearning: A survey” (Liu et al., 2024) ), amongst other concerns requiring changes to datasets. In the context of the unlearning literature, this paper tackles an important and novel problem of providing an algorithm with unlearning guarantees in novel settings (however claims need to be clarified).

**Theoretical Claims:**

See claims and evidence for issues with the unlearning definition and how it then presents inconsistencies with past work.

This said, I believe the proofs are correct given the training algorithm ends with $T$ steps of clipped (either gradient or model) noisy fine-tuning on the training dataset. I checked the main proofs of Theorem 4.1 and 4.2.

---

> ### Author Rebuttal · Authors · 2025-04-01
>
> Thank you for your time and valuable comments that will allow us to improve our manuscript.
>
>
> ### **Clarification on Definition 2.1:**
>
> We apologize for the confusion caused. We believe that the reviewer misinterpreted our definition. We would like to note that **no modification to the training algorithm is required in our work**.
>
> Our Definition 2.1 is intended as a general and unifying framework that covers multiple previous definitions of unlearning. We want to clarify that in Definition 2.1. we did not predefine the certifying algorithm $\bar A$ for generality. We only ask for the existence of such a certifying algorithm $\bar A$ and its independence from the dataset $D_f$. Specifically, our definition captures:
>
> * (Ginart et al., 2019; Guo et al., 2019), where the certifying algorithm $\bar A$ equals the training algorithm $A$.
> * (Sekhari et al., 2021; Allouah et al., 2024), where $\bar A(.) = U(A(.), . ,∅)$.
>
> In our work we provide certification with $\bar A(.) = U(A(.), . ,\varnothing)$. A similar choice was made in several prior works, such as (Sekhari et al, 2021) and (Allouah et al, 2024), and therefore is not a novelty of our work. Such a choice allows us to prove the unlearning guarantees of our algorithms (3) and (4).
>
> Importantly, our choice of the certifying algorithm $\bar A$ is purely theoretical and does not require running additional computational steps in practice. Thus, there is no practical modification of the original training required. Consequently, comparisons with methods that fundamentally modify training (e.g., SISA or naive DP training) fall outside our experimental setup.
> We appreciate your suggestion about explicitly clarifying these points in the manuscript. We will revise Definition 2.1 and associated discussions clearly stating how our definition captures prior definitions and the choice of the certifying algorithm $\bar A$.
>
> ### **About Thudi et al. (2022):**
>
> We would like to thank the reviewer for pointing to the related work. We will add it to the next version of our paper.
>
> Considering the adversarial setting with server possibly forging unlearning as was done in (Thudi et al, 2022) is an interesting problem but beyond the scope of our work. In our work, we focus on developing the algorithms for how to achieve unlearning in the non-adversarial (honest server) setting, a problem that has previously remained unsolved for general non-convex functions. Providing a certification algorithm for our approach in the adversarial setting is an interesting direction for future work but orthogonal to current work.
>
> ### **Responses to Specific Questions:**
>
> 1. **Accuracy on CIFAR-10 and MNIST:** Our reported accuracy is indeed lower than typical ResNet results because we employed much smaller neural networks. Higher-dimensional settings suffer performance degradation similar to standard DP-training methods. We will highlight this in the paper, and we acknowledge that future experiments with larger, state-of-the-art architectures are certainly valuable.
>
>
> 2. **Hyperparameter selection:** The hyperparameters were tuned based on standard grid search procedures to balance performance guarantees, while strictly following the theoretical guidelines relating the parameters (e.g., Theorem 4.1 indicates the noise magnitude given the number of iterations). We list the result of our grid search in Appendix B. We will provide explicit explanations in the revised manuscript.
>
> We hope that we have successfully clarified any confusion regarding our definition 2.1 and addressed the concerns raised by the reviewer. If this is the case, we kindly ask the reviewer to reconsider their evaluation and raise their score accordingly.

---

> > ### Comment · Reviewer_z3Fs · 2025-04-02
> >
> > Thank you for your response!
> >
> > I now understand the context for the definition, and thank the authors for their detailed response. To summarize my current understanding of the main contribution of the paper, this paper shows that for this weaker "post-processing" unlearning definition (as in methods satisfying Ginart et al., also satisfy this definition, but not vice-versa) we do not need to put restriction on the original training algorithm. I am raising my score given this clarification. However, I emphasize again that the authors should explicitly clarify in the paper that this is a weaker unlearning definition than considered in a large part of the literature. Importantly, a now open problem is understanding use-cases for this "post-processing" unlearning (e.g., when someone asks for their data to be removed, is it enough for it to look like what we would have post-processed if it was not in the dataset?).
> >
> > Also, I still find an issue with the evaluation is that DP-SGD and SISA are baseline methods satisfying this definition, neither of which are compared to. The authors argue DP-SGD and SISA approaches fall outside the scope of their paper as they modify the training algorithm. However, I find the current experiment suite does not answer what we gain/lose by doing this post-processing method (relative to approaches that do change the learning algorithm/are not post-processing). The authors mention they expect to suffer similar trade-offs as DP-SGD, and will discussion this as future work. I wish to emphasize that SISA would seemingly not suffer such a trade-off, and so the (practical) benefits of this method compared to it are less clear beyond not modifying the training algorithm (but this seems a weak statement as one still needs to post-process and lose performance). Ultimately having the experiments to quantitatively show the limits of this method would strengthen the paper, in my opinion.

---

> > > ### Author Response · Authors · 2025-04-09
> > >
> > > We thank the reviewer for their thoughtful feedback. To address concerns around evaluation in more complex settings, we conducted **new experiments** on CIFAR-100 and CIFAR-10 using **ResNet architectures** pretrained on public data (ImageNet). This setup, where unlearning is applied to the last few layers of a pretrained model, has become standard in recent certified approximate unlearning works (e.g., Guo et al. 2020, Chien et al. 2024). However, prior works restrict themselves to convex settings (i.e., linear final layer), whereas our method is **the first to provide certified unlearning guarantees for multiple non-convex layers, without any smoothness/convexity assumptions.**
> > >
> > > More precisely, we remove the last layer of ResNet-18 (pretrained on public data) and replace it with a 3-layer fully connected neural network. We first train the last 3 layers of our resulting architecture on the full data, and then unlearn the forget data from these 3 layers.
> > >
> > > To demonstrate practical effectiveness, we compare our method against **DP-SGD (ε = 50)**, as suggested by reviewers, and **retraining**, while maintaining a much stricter ε = 1 guarantee. DP-SGD enforces privacy before unlearning, followed by additional fine-tuning during unlearning.
> > >
> > > As shown below, similar to Table 2 in the paper, our method consistently requires fewer epochs-- up to **2–3× less compute** than DP-SGD, and faster than retraining in high-accuracy regimes. The tables report the number of training epochs needed to reach each target accuracy.
> > >
> > > ### **CIFAR-100**
> > >
> > > | Accuracy   |   Gradient Clipping (ours) | Retrain            | DP-SGD              |
> > > |:-----------|--------------------:|:-------------------|:--------------------|
> > > | 50%        |                   14 | 17 (≈ 18% slower)   | >50 (> 72% slower) |
> > > | 53%        |                   18 | 20 (≈ 10% slower)   | >50 (> 54% slower) |
> > > | 55%        |                   20 | 22 (≈ 9% slower)  | >50 (>60% slower) |
> > > | 58%        |                   26 | 29 (≈ 10 % slower)   | >50 (>48% slower)|
> > > | 60%        |                   32 | 34 (≈ 6% slower)   | >50 (>36% slower)  |
> > > | 62%        |                   39 | >50 (> 22 %slower)  | >50 (> 22 %slower) |
> > >
> > >
> > > ### **CIFAR-10**
> > >
> > > | Accuracy   | Gradient Clipping (ours)   | Retrain            | DP-SGD             |
> > > |:-----------|:--------------------|:-------------------|:-------------------|
> > > | 85%        | 9                   | 10 (≈ 10 % slower)   | 17 (≈ 47 % slower)   |
> > > | 86%        | 14                   | 17 (≈ 18 % slower)   | 22 (≈ 36 % slower)   |
> > > | 87%        | 21                   | 28 (≈ 25 % slower)   | 35 (≈ 40 % slower)   |
> > > | 88%        | 39                   | >50 (> 22% slower)   | >50 (> 22 % slower)   |
> > >
> > > These results demonstrate that our method achieves significant gains in both privacy and efficiency, even outperforming DP-SGD under a much tighter certificate $\epsilon$. Crucially, we do so **without modifying the original training process**, placing our work in the post-processing regime.
> > >
> > > Finally, we view our method as orthogonal to SISA. It could be used to improve shard retraining when approximate unlearning is sufficient, though combining both directions is beyond the scope of this work.
> > >
> > > While certified unlearning in non-convex settings remains an open challenge, we believe this work represents a **major step forward**, bridging the gap between formal guarantees and practical applicability in deep learning. We hope that these additional results and clarifications effectively address your concerns, and we would greatly appreciate it if you would consider raising your score based on this.

---

### Official Review · Reviewer_GtAL · 2025-03-11

**Overall Recommendation:** 3

**Summary:**

The paper presents a post processing technique to guarantee approximate unlearning in non-convex settings. The paper builds on several works in the differential privacy literature that incorporated noise during the optimization process to improve privacy guarantees. The paper proposes two methods: gradient clipping and model clipping to achieve unlearning. The paper theoretically computes the degree of noise required in each setting as a function of clipping and number of iterations. Further results show that the proposed method requires a significantly smaller amount of noise per iteration compared to the baselines.

**Claims And Evidence:**

Strengths:


- The paper is well motivated and very easy to follow.

- The paper proposes a simple method for approximate unlearning backed by theoretical guarantees that extend to non-convex settings. This removes the need to know several characteristics of the existing loss function like the smoothness constant.

**Essential References Not Discussed:**

NA

**Experimental Designs Or Analyses:**

NA

**Methods And Evaluation Criteria:**

Weaknesses:

- The evaluation of the paper seems limited as the experiments only focus on MNIST and CIFAR10 datasets, and use very small scale neural networks. I would like to see results using larger models and more complex datasets. Specifically, I'm interested to know if the updates could be applied to a subset of the model parameters to achieve $(\epsilon, \delta)$ unlearning.

- An obvious baseline that I would like to see is DP-SGD. What if we trained the model from scratch using differential privacy? How does the unlearning performance compare with DP-SGD?

- The paper should report more experiments using different levels of $\epsilon$. Report the final accuracy and the number of iterations required for different $\epsilon$.

- In Table 2, we observe that the proposed method requires a large number of fine-tuning steps that is comparable to retraining the model from scratch. How does this compare with exact unlearning systems? These systems reduce the unlearning steps by using modular systems and can retain high performance. I understand that this is a different paradigm of unlearning, but it would be interesting to discuss and compare the shortcomings of each unlearning technique.


[1] https://arxiv.org/abs/1912.03817

[2] https://arxiv.org/pdf/2406.16257

**Other Comments Or Suggestions:**

NA

**Other Strengths And Weaknesses:**

NA

**Questions For Authors:**

Please respond to the weaknesses above.

**Relation To Broader Scientific Literature:**

NA

**Theoretical Claims:**

The theoretical claims in the paper look good to me. However, I'm not an expert in differential privacy and the proofs of the theoretical claims are beyond the scope of my expertise.

---

> ### Author Rebuttal · Authors · 2025-04-01
>
> We thank the reviewer for their time and constructive feedback.
>
> 1. We acknowledge that our experiments currently focus on fundamental settings (MNIST, CIFAR10), yet our primary contribution lies in providing rigorous theoretical guarantees without restrictive assumptions such as smoothness or convexity.
>
> 2. DP-SGD changes the training algorithm fundamentally and, therefore, does not fit into our setting. Moreover, if we were to implement the DP-SGD baseline, we would need the per iteration noise to be scaled linearly with the forget set size, compared to the standard DP training. Since our forget set size is 5000 for CIFAR-10 and 6000 for MNIST, we do not expect any good practical performance. There is also theoretical evidence that unlearning for free via DP is severely limited in high dimensions (Allouah et al, 2024).
>
> 3. Including additional experiments varying $\varepsilon$ is a great suggestion. We will include such results clearly in our revised paper. We do not expect any significant difference in the relative performance.
>
> 4. We appreciate the suggestion regarding exact unlearning systems, which indeed modify the training algorithm. We are unaware of any exact unlearning system that does not modify original training, except for retraining from scratch. We will add a discussion on this alternative approach to the related work of our paper. The exact unlearning systems typically modify the training, losing the performance quality of the model e.g., SISA (Bourtoule et al., 2019), however, they might be more efficient when unlearning.

---

### Official Review · Reviewer_THT1 · 2025-03-14

**Overall Recommendation:** 3

**Summary:**

The paper proposes a novel certified unlearning method that integrates noisy fine-tuning with privacy amplification by stochatic post-processing, which introduces gradient clipping and model clipping, both combined with Gaussian privacy noise. The authors provide rigorous theoretical analysis for unlearning guarantees that do not depend on restrictive assumptions. Empirical results demonstrate the effectiveness of the proposed method.

**Claims And Evidence:**

Yes.

**Essential References Not Discussed:**

The paper focuses on the field of certified machine unlearning. However, it only cites certified unlearning methods under the conventional convex setting while missing the works for other specific settings, e.g., graph neural networks [1] and minimax models [2].

[1] Eli Chien, Chao Pan, and Olgica Milenkovic. “Certified Graph Unlearning”.

[2] Jiaqi Liu, Jian Lou, Zhan Qin, and Kui Ren. “Certified Minimax Unlearning with Generalization Rates and Deletion Capacity”. In NeurIPS 2023.

**Experimental Designs Or Analyses:**

The reviewer has checked all of the experimental designs.

**Methods And Evaluation Criteria:**

Yes.

**Other Comments Or Suggestions:**

1. Lack of explanation of notations ($C_0, C_1, C_2, \lambda$ and $\gamma$) in the table caption.

**Other Strengths And Weaknesses:**

Strengths:

1. The unlearning guarantees do not depend on restrictive assumptions such as loss function smoothness. The theoretical analysis is rigorous.
2. The paper is overall well-structured. The narrative is easy to follow.

Weaknesses:

1. The network used in the experiments is simple, and the datasets are small. Although this paper mainly focuses on the theoretical part, it would be better to include the experimental results on larger datasets, e.g., ImageNet.

**Questions For Authors:**

It seems that the regularization factor $\gamma$ does not affect the unlearning guarantee results of model clipping. Please explain the role of $\gamma$ here.

**Relation To Broader Scientific Literature:**

This paper presents work whose goal is to advance the field of machine unlearning, which is specifically oriented to improve the trustworthiness of machine learning.

**Theoretical Claims:**

The reviewer did not check the correctness of the proofs.

---

> ### Author Rebuttal · Authors · 2025-04-01
>
> We thank the reviewer for their time and constructive feedback.
>
> 1. Thanks for pointing to the papers about unlearning for graph neural networks and minmax models. We will add these references to the related works section.
>
> 2. We acknowledge that our experiments currently focus on fundamental settings (MNIST, CIFAR10), yet our primary contribution lies in providing rigorous theoretical guarantees without restrictive assumptions such as smoothness or convexity.
>
> 3. We will add the explanation of notations $C_0,C_1,C_2, \lambda$, and $\gamma$ to the table caption.
>
> 4. Regularization factor $\gamma$: Thank you for raising this question. The regularization factor $\gamma$ plays a crucial role in our theoretical analysis, related to privacy amplification via iteration (see Sec. 4.1), as the required noise magnitude decreases exponentially in the number of iterations $T$ thanks to regularization. In practice, it helps control the norm of the model parameters throughout fine-tuning, which directly influences the amount of noise required per iteration to ensure certified unlearning guarantees. We will clarify this role of $\gamma$ explicitly in the revised manuscript.

---

### Official Review · Reviewer_7Qas · 2025-03-17

**Overall Recommendation:** 3

**Summary:**

**Main Results**: Although the idea of fine-tuning an originally trained model on retained data has been proposed before, it has traditionally been viewed as an empirical forgetting strategy for non-convex tasks. This paper provides certified unlearning guarantees for neural networks without requiring knowledge of the smoothness constant of the loss function.

**Main Algorithmic/Conceptual Ideas**: The authors propose two clipping-based strategies: gradient clipping and model clipping. The core idea involves either "clip-before-updating" or "update-before-clipping," with the addition of Gaussian noise to the output.

**Main Findings**: The authors provide approximate unlearning guarantees for both methods. Then they compare the performance of their methods with the baseline (output perturbation) when achieving the same unlearning guarantee.

## update after rebuttal

The authors have addressed most of my concerns by conducting additional empirical evaluations, and the results looks reasonable. Considering the comments and responses from the other reviewers, I have decided to raise my score to "weak accept."

**Claims And Evidence:**

The authors offer detailed proofs to support the certified unlearning guarantees of their proposed methods, and upon review, these proofs appear sound and logical.

However, the experimental analyses presented are somewhat basic and do not sufficiently explore or demonstrate the effectiveness/feasibility of the proposed approaches in more complicated scenarios. See more details in *Methods And Evaluation Criteria* and *Experimental Designs Or Analyses*.

**Essential References Not Discussed:**

It appears that the authors have overlooked a highly relevant paper that addresses a similar problem:

[1] Binchi Zhang, Yushun Dong, Tianhao Wang, Jundong Li. "Towards Certified Unlearning for Deep Neural Networks." ICML 2024.

**Experimental Designs Or Analyses:**

+ The authors primarily compare their methods to output perturbation and the most naive baseline method, 'retrain.' However, it would be beneficial to include comparisons with other related methods that also provide certified unlearning guarantees, such as using Newton updates [1], Fisher forgetting [2], and state-of-the-art methods proposed in [3] and [4].

+ The authors do not conduct experiments in more practical settings to verify the feasibility of their proposed methods. For example,
    - the sequential setting where users can send unlearning requests at different time points in sequence
    - the microbatch deletion setting where the size of the forget set varies, such as 0.1%, 1%, 10%, etc.

+ Regarding the statement on page 7: “the privacy target is reached before exhausting the iteration budget, in less than 100 iterations,” the terms “privacy target” and “iteration budget” are somewhat confusing. It seems these terms have not been clearly defined earlier in the text.
    - What is the relationship between these terms and the “accuracy target” and “compute budget” mentioned later?
    - Does the "privacy target" refer to the “(ε, δ)-unlearning guarantee,” and does the "iteration budget" refer to the number of iterations required to reach the target accuracy?

+ In Figure 2, if I understand correctly, the accuracy improvement seems to primarily result from standard fine-tuning on the retained set, as the accuracy after the noisy steps drops to nearly zero. Moreover, from the convergence curve, it appears that this "first-noisy-then-standard" fine-tuning procedure only results in less than a 0.1 accuracy improvement compared to retraining from scratch, however, with a similar computational cost and more significant fluctuations (especially during the early unlearning epochs) . The authors are expected to provide more justifications for those findings.




References:\
[1] Certified Data Removal from Machine Learning Models. \
[2] Golatkar, A., Achille, A., Ravichandran, A., Polito, M., and Soatto, S. (2021). Mixed-privacy forgetting in deep networks. ICCV. \
[3] Youssef Allouah, et al. "The Utility and Complexity of In- and Out-of-Distribution Machine Unlearning." ICLR 2025. \
[4] Binchi Zhang, Yushun Dong, Tianhao Wang, Jundong Li. "Towards Certified Unlearning for Deep Neural Networks." ICML 2024.

**Methods And Evaluation Criteria:**

The evaluation criteria include: 1) the number of steps required to achieve a fixed target accuracy, and 2) the validation accuracy attained when the number of update steps is fixed.

To strengthen their analysis, the authors could provide theoretical justifications regarding the utility and complexity trade-offs, as discussed in [1]. Alternatively, they could offer empirical justifications focusing on relearn time, the accuracy of membership inference attacks (MIA), and the AUC score of MIA, as explored in [2].

[1] Youssef Allouah, et al. "The Utility and Complexity of In- and Out-of-Distribution Machine Unlearning." ICLR 2025.
[2] Binchi Zhang, Yushun Dong, Tianhao Wang, Jundong Li. "Towards Certified Unlearning for Deep Neural Networks." ICML 2024.

**Other Comments Or Suggestions:**

A Minor Comment:

The authors use several terms—"epoch," "iteration," "time," "compute budget," and "iteration budget"—that seem to convey similar meanings. Could the authors clarify the distinctions between these terms?

**Other Strengths And Weaknesses:**

**Originality**: The problem of establishing a certified unlearning guarantee for fine-tuning-based methods in non-convex cases is well-motivated.

**Clarity**: As noted in the previous comments, the paper uses somewhat vague language and lacks sufficient theoretical or empirical justification for its methods. Many unclear statements should be clarified in the revised version. Refer to [Questions for Authors] X for specific areas needing improvement.

Significance: I think the methods proposed in this paper could be of interest to the machine unlearning community.

**Questions For Authors:**

The authors are expected to address all previously mentioned issues, especially the sections on *Methods and Evaluation Criteria* and *Experimental Designs or Analyses.*

**Relation To Broader Scientific Literature:**

N/A

**Theoretical Claims:**

I reviewed the proofs of Theorem 4.1 and Theorem 4.2, and they seemed logical and coherent to me. However, I must admit that I did not examine their correctness in great detail.

---

> ### Author Rebuttal · Authors · 2025-04-01
>
> We thank the reviewer for the detailed and constructive feedback, and we address the key points below.
>
>
> **1. Baseline Comparisons:**
>
> We choose only output perturbation and retraining from scratch since these algorithms are the only baselines in the literature that can achieve certified unlearning without additional assumptions on the loss function or modification of the original training.
> In more details:
> * Newton updates [1]: assume **convexity and smoothness** of the loss function, therefore, are inapplicable for the unlearning deep neural networks.
> * Golatkar et al [2]: impose the **smoothness assumption** on the loss function, which frequently does not hold in deep learning if activation functions are non-smooth. Moreover, their unlearning definition is based on mutual information, and it is non-trivial how to connect it with our differential privacy-based definition of unlearning.
> * Allouah et al [3]: impose additional assumptions of either **strong convexity and smoothness** or assume **smoothness and that the minima is unique and achievable from any initialization**. Such assumptions do not hold in the deep learning setting.
> * Zhang et al. [4]: require the loss function to be **L-smooth, as well as the knowledge of the minimal eigenvalue of the hessian**, limiting its applicability to the general deep learning setting.
>
> We are unaware of any other work that can tackle certified unlearning in the same generality as our work without limiting additional assumptions on the loss functions.
>
> We will add this discussion to the next version of our manuscript.
>
> **2. Utility-complexity tradeoff:**
>
> Thank you for suggesting a discussion of utility-complexity tradeoffs as explored in [1]. However, such tradeoffs typically require strong assumptions such as convexity or smoothness. Given our primary contribution is providing guarantees for general non-convex settings without these assumptions, clearly characterizing utility-complexity tradeoffs becomes inherently challenging—particularly due to issues like the curse of dimensionality and non-smoothness. We will clarify this inherent difficulty in our revised manuscript.
>
>
> **3. Empirical Justifications (Accuracy and AUC of MIA):**
>
> Our goal is to provide rigorous theoretical guarantees without imposing restrictive assumptions such as smoothness or knowledge of eigenvalues of Hessian matrices, as required by [2]. Hence, empirical measures such as MIA scores, used to evaluate methods without strong theoretical guarantees, are not directly comparable or necessary in our theoretical setting. Nevertheless, connecting theoretical unlearning guarantees to practical empirical metrics is an intriguing direction for future exploration, but beyond the scope of this work.
>
> **4. Accuracy Improvements (Fig. 2):**
>
> Our experimental results (Table 2, Figure 1) show consistent improvement over retraining from scratch across all compute budgets and accuracy targets. Notably, Figure 1 illustrates that with a small compute budget, our method improves test accuracy by up to 8%. The improvement is most significant in low-compute settings, while larger compute budgets make retraining from scratch more effective. In Figure 2, where the compute budget is relatively large, the accuracy improvement is less pronounced. Importantly, both our method and retraining from scratch employ fine-tuning on the retained set. Our approach preserves useful information from the trained model, making fine-tuning more effective.
>
> **5. Experimental Setup and Practical Settings:**
>
> We agree that extending our experimental framework (e.g., sequential or microbatch deletion settings, as suggested by the reviewer) would provide additional practical insights. While our current focus remains on fundamental theoretical generality, we recognize the value of practical validation and will explicitly outline these as future directions.
>
> **6. Terminology Clarification:**
>
> We acknowledge the confusion caused by unclear terminology ("privacy target," "iteration budget," etc.). Yes, your interpretations are correct—the "privacy target" refers to the "$(\varepsilon, \delta)$-unlearning guarantee," while the "iteration budget" corresponds to the maximum allowed iterations. We will clearly define these terms in the revised manuscript.
>
>
>
>
>
> ### References
>
> [1] Guo et al. Certified Data Removal from Machine Learning Models. ICML 2020.
>
> [2] Golatkar et al. Mixed-privacy forgetting in deep networks. ICCV 2021.
>
> [3] Allouah et al. "The Utility and Complexity of In- and Out-of-Distribution Machine Unlearning." ICLR 2025.
>
> [4] Zhang et al. "Towards Certified Unlearning for Deep Neural Networks." ICML 2024.

---

> > ### Comment · Reviewer_7Qas · 2025-04-02
> >
> > Dear authors,
> >
> > Thank you for your response. I appreciate your efforts to address the review comments.
> >
> > However, I find that the rebuttal does not sufficiently address my initial concerns and provides compelling new evidence to alter my assessment. To strengthen the manuscript, I still recommend implementing **at least one** of the following improvements:
> >
> > 1. Conduct a more comprehensive empirical evaluation (following the experimental setups of the closely related work [1]);
> >
> > 2. Provide formal analysis of either utility guarantees or computational complexity.
> >
> > After reading the rest of reviews and the responses, I decided to maintain my initial rating.
> >
> >
> > [1] Binchi Zhang, Yushun Dong, Tianhao Wang, Jundong Li. "Towards Certified Unlearning for Deep Neural Networks." ICML 2024.

---

> > > ### Author Response · Authors · 2025-04-09
> > >
> > > We thank the reviewer for their thoughtful feedback. To address concerns around evaluation in more complex settings, we conducted **new experiments** on CIFAR-100 and CIFAR-10 using **ResNet architectures** pretrained on public data (ImageNet). This setup, where unlearning is applied to the last few layers of a pretrained model, has become standard in recent certified approximate unlearning works (e.g., Guo et al. 2020, Chien et al. 2024). However, prior works restrict themselves to convex settings (i.e., linear final layer), whereas our method is **the first to provide certified unlearning guarantees for multiple non-convex layers, without any smoothness/convexity assumptions.**
> > >
> > > More precisely, we remove the last layer of ResNet-18 (pretrained on public data) and replace it with a 3-layer fully connected neural network. We first train the last 3 layers of our resulting architecture on the full data, and then unlearn the forget data from these 3 layers.
> > >
> > > To demonstrate practical effectiveness, we compare our method against **DP-SGD (ε = 50)**, as suggested by reviewers, and **retraining**, while maintaining a much stricter ε = 1 guarantee. DP-SGD enforces privacy before unlearning, followed by additional fine-tuning during unlearning.
> > >
> > > As shown below, similar to Table 2 in the paper, our method consistently requires fewer epochs-- up to **2–3× less compute** than DP-SGD, and faster than retraining in high-accuracy regimes. The tables report the number of training epochs needed to reach each target accuracy.
> > >
> > > ### **CIFAR-100**
> > >
> > > | Accuracy   |   Gradient Clipping (ours) | Retrain            | DP-SGD              |
> > > |:-----------|--------------------:|:-------------------|:--------------------|
> > > | 50%        |                   14 | 17 (≈ 18% slower)   | >50 (> 72% slower) |
> > > | 53%        |                   18 | 20 (≈ 10% slower)   | >50 (> 54% slower) |
> > > | 55%        |                   20 | 22 (≈ 9% slower)  | >50 (>60% slower) |
> > > | 58%        |                   26 | 29 (≈ 10 % slower)   | >50 (>48% slower)|
> > > | 60%        |                   32 | 34 (≈ 6% slower)   | >50 (>36% slower)  |
> > > | 62%        |                   39 | >50 (> 22 %slower)  | >50 (> 22 %slower) |
> > >
> > >
> > > ### **CIFAR-10**
> > >
> > > | Accuracy   | Gradient Clipping (ours)   | Retrain            | DP-SGD             |
> > > |:-----------|:--------------------|:-------------------|:-------------------|
> > > | 85%        | 9                   | 10 (≈ 10 % slower)   | 17 (≈ 47 % slower)   |
> > > | 86%        | 14                   | 17 (≈ 18 % slower)   | 22 (≈ 36 % slower)   |
> > > | 87%        | 21                   | 28 (≈ 25 % slower)   | 35 (≈ 40 % slower)   |
> > > | 88%        | 39                   | >50 (> 22% slower)   | >50 (> 22 % slower)   |
> > >
> > > These results demonstrate that our method achieves significant gains in both privacy and efficiency, even outperforming DP-SGD under a much tighter certificate $\epsilon$. Crucially, we do so **without modifying the original training process**, placing our work in the post-processing regime.
> > >
> > > We also thank the reviewer for pointing out the relevance of Zhang et al., ICML 2024, and we will include a proper citation and discussion of this work, similar to what we outlined in our initial rebuttal, in the revised manuscript.
> > >
> > > While certified unlearning in non-convex settings remains an open challenge, we believe this work represents a **major step forward**, bridging the gap between formal guarantees and practical applicability in deep learning. We hope that these additional results and clarifications effectively address your concerns, and we would greatly appreciate it if you would consider raising your score based on this.

---

### Decision · Program_Chairs · 2025-05-01

**Decision:**

Accept (poster)

**Comment:**

This paper studies the problem of developing provable unlearning algorithms for neural networks by leveraging ideas from differential privacy. The basic idea is that you clip gradients before applying gradient updates (but only to the retain set) and then add noise. An alternative is where the model parameters are clipped prior to adding noise. In both cases the algorithms are proven to satisfy approximate unlearning guarantees. The claimed technical innovation beyond prior work is the relaxation of the requirement of convexity and smoothness functions in the analysis step. During the rebuttal the authors clarified with the reviewers on where the work is situated with respect to the varying definition of unlearning (z3Fs) and provided additional empirical evaluations on CIFAR10 and CIFAR100 (z3Fs, 7Qas). I think its important to include the above empirical results and discussions within the manuscript. In addition, from my reading of the discussion, the response to reviewer 7Qas does a good job of highlighting how the theoretical asssumptions of proving (eps-del) unlearning here were distinct from Guo, Golatkar, Allouah and Zhang's work, but it does a poor job of reckoning with the lack of empirical validation on how much of a difference the relaxation of smoothness and convexity actually makes in practice. i.e. is there a continuum of tradeoffs in algorithms that relax theoretical performance in approximate unlearning with empirical performance? I think discussing what this manuscript does not do in the discussion would further strengthen directions for future work.